# Eigenmode orthogonality breaking and anomalous dynamics in multimode nano-optomechanical systems under non-reciprocal coupling

Laure Mercier de Lépinay[1], Benjamin Pigeau[1], Benjamin Besga[1] & Olivier Arcizet[1]

Thermal motion of nanomechanical probes directly impacts their sensitivities to external forces. Its proper understanding is therefore critical for ultimate force sensing. Here, we investigate a vectorial force field sensor: a singly-clamped nanowire oscillating along two quasi-frequency-degenerate transverse directions. Its insertion in a rotational optical force field couples its eigenmodes non-symmetrically, causing dramatic modifications of its mechanical properties. In particular, the eigenmodes lose their intrinsic orthogonality. We show that this circumstance is at the origin of an anomalous excess of noise and of a violation of the fluctuation dissipation relation. Our model, which quantitatively accounts for all observations, provides a novel modified version of the fluctuation dissipation theorem that remains valid in non-conservative rotational force fields, and that reveals the prominent role of non-axial mechanical susceptibilities. These findings help understand the intriguing properties of thermal fluctuations in non-reciprocally-coupled multimode systems.

[1] Institut Néel, Université Grenoble Alpes - CNRS:UPR2940, 38042 Grenoble, France. Correspondence and requests for materials should be addressed to O.A. (email: olivier.arcizet@neel.cnrs.fr)

The sensitivity of any sensor based on a mechanical oscillator is intrinsically limited by its thermal noise, a random position fluctuation which can mask the signal under investigation. Hence, understanding and reducing thermal noise is a permanent objective in force, position or metric sensing to approach ultimate sensitivities.

The recent advent of nanomechanical oscillators[1–3] has boosted the force sensitivity by orders of magnitudes[4], enabling single electron imaging[5], or ultimate mass sensing[6–8], opening perspectives in both fundamental and applied science[9–13]. Their ultralow-stiffness generally comes with a sizeable thermal position noise, that can spread over distances approaching their intrinsic dimensions. These resonators are all the more sensitive to environmental inhomogeneities which manifest themselves as external force field gradients. The latter can be mapped by tracking modifications of the probe nanomechanical properties (frequency shifts, change in oscillation amplitude, or damping rates) as standardly used in atomic force microscopes[14]. The force gradients are probed along the oscillating direction, either normal or parallel[15] to the sample. Consequently, a force probe deformable along several directions of space allows for vectorial force field imaging[16–18], conveying a supplementary physical richness to a traditionally scalar measurement. In particular multidimensionality allows for force fields to present vorticity. Therefore it becomes possible to inspect the role of non-conservative force landscapes, that is, force fields that do not derive from a potential energy, here because they sustain a non-zero rotational.

Our 2D nanomechanical force probe is a singly clamped suspended nanowire oscillating along both transverse directions[13, 16–25]. Nanoresonators with similar geometry were employed in MRFM applications[20, 26], in electrostatic force field sensing[17–19], but all the developments presented here also apply to doubly clamped nanobeams oscillating in and out of plane[4, 27–30] and more generally to multimode mechanical systems coupled through any external force field. Nanowires identical to the ones used in this work were previously employed to map the optical force field generated at the waist of a strongly focused laser beam, using pump-probe measurements at low power in absence of distortion of the eigenmodes basis. This revealed the non-conservative nature of the optical force field through a direct measurement of its non-zero force rotational[16]. It was theoretically suggested that it could generate a warping of the eigenmode basis and an altered dynamics, but no experimental proof could be drawn since only a scalar 1D readout was available at that time. Recently, a novel universal method based on a dual, 2D motion detection was introduced[17] to image any 2D force fields even when pump-probe measurements[16, 18] cannot be employed. Simultaneous tracking of the eigenfrequency shifts and eigenmode rotations of a quasi frequency-degenerate nanowire gives access to the entire structure of the 2D force field gradient experienced by the nanowire extremity. This method is immune to a distortion of the eigenmode basis and therefore not limited to the measurement of small force fields gradients. In particular it gives access to shear components of the force field which are essential to understand the physics at play in sharply varying force fields landscapes. The measurement principle was verified on a conservative electrostatic force field[17]. The present work proves and extends its validity in the case of non-conservative force fields.

Non-conservative force fields can couple mechanical degrees of freedom in a non-symmetric way. For example, if a charged particle is immersed in a magnetic field, the components of its motion along perpendicular directions are non-symmetrically coupled through the Lorentz force: the force experienced along the two transverse directions being proportional to the speed along the other direction but with an opposite sign. Similarly, the Coriolis force, which is also proportional to the speed of a moving mass, is responsible for the apparent rotation of a Foucault pendulum. As early noticed in optical tweezers experiments[31–33], the radiation pressure force field is non-conservative as it presents a non-zero rotational. It was shown that this optical force field structure can strongly alter the Brownian motion of an over-damped trapped particle, causing the emergence of so-called Brownian vortexes[34, 35]. Here, we exploit an analogous rotational optical coupling force, but on an under-damped resonator which permits to investigate eigenmodes cross-coupling. This arrangement thus represents a unique situation where non-symmetric but reactive (instantaneous) mode coupling is achieved via a rotational force field. The field of cavity optomechanics has recently developed a strong interest in multimode coupling phenomena such as back-action cancelation[36], two-mode back-action-evading measurements[37], cavity-assisted optical hybridization between mechanical eigenmodes[38–41] and this class of systems was also subject to significant theoretical developments[42–44]. In those implementations, only one single cavity mode was involved, so that the cavity-mediated mechanical cross-coupling remained symmetric (even though non-reciprocal energy transfer[41] could be achieved on proper paths in control parameter space). Asymmetric mechanical coupling should in principle arise in multimode cavity optomechanics using multiple optical modes[45, 46] or engineered optomechanical arrays[47].

In this work, we investigate the thermal noise and the driven dynamics of a singly clamped suspended nanowire oscillating along both transverse directions immersed in a optical tunable rotational force field. We report on the observation of the warping of the eigenmodes basis, thus breaking its original orthogonality, and on an alteration of the thermal noise spectra. We show experimentally that the position fluctuations now deviate from the fluctuation dissipation relation (FDR) in its original formulation, as the rotational force field brings the system out of equilibrium, which violates a fundamental hypothesis used for the FDR derivation. All these rather intriguing observations are quantitatively explained by our model which also suggests a generic methodology to correctly describe the fluctuations and dynamical properties of multimode resonators strongly coupled by non-reciprocal mechanisms. Our experiment thus represents a first exploration of the novel physics emerging in this non-reciprocal coupling situation, in a simple system where all discussed quantities can be directly mapped and visualized in a 2D space.

## Results

**The experiment**. The experiment is sketched in Fig. 1a. It is conducted on a 165 μm-long and $\simeq$120 nm-diameter Silicon Carbide nanowire (NW) suspended at the extremity of a sharp tungsten tip. Using high numerical aperture microscope objectives and a XYZ piezostage, a 633 nm probe laser is strongly focused on the NW vibrating extremity. Its position fluctuations $\delta\mathbf{r}(t)$ are encoded on the reflected field. This field is collected on a split photodetector followed by a low noise amplifier providing the sum and difference of photovoltages: $V_{\ominus,\oplus}(\mathbf{r_0} + \delta\mathbf{r}(t))$. Their temporal fluctuations $\delta V_{\ominus,\oplus}(t) = \delta\mathbf{r}(t) \cdot \boldsymbol{\nabla} V_{\ominus,\oplus}$ convey projective measurements of the 2D NW trajectory, $\delta r_{\ominus,\oplus} \equiv \delta\mathbf{r}(t) \cdot \mathbf{e}_{\ominus,\oplus}$, projected along two measurement vectors $\mathbf{e}_{\ominus,\oplus} \equiv \boldsymbol{\nabla} V_{\ominus,\oplus}/|\boldsymbol{\nabla} V_{\ominus,\oplus}|$, which are simply the local gradients of the maps shown in Fig. 1b. Operating at the working point highlighted by $\odot$ in Fig. 1b where the measurement vectors are quasi-orthogonal permits to realize a full 2D readout of the NW position fluctuations through a simultaneous acquisition of both signals. A position tracking can be activated to stabilize the NW position with respect to the probe

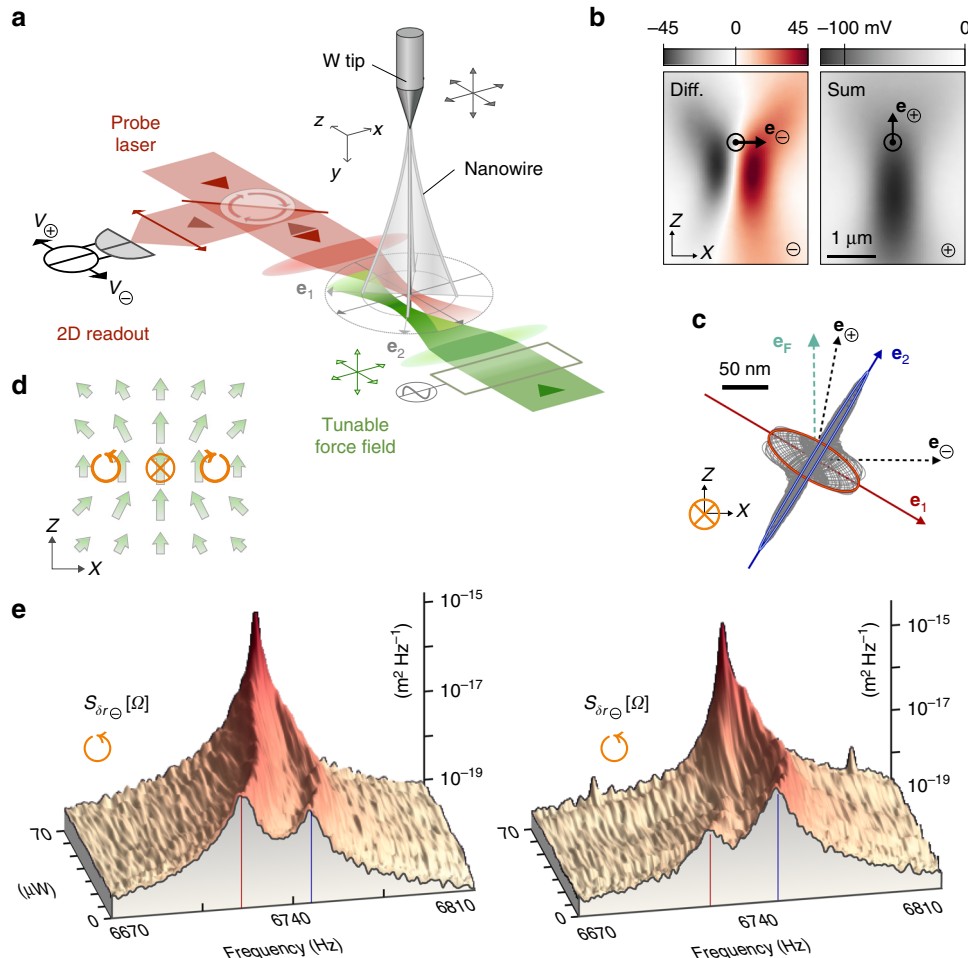

**Fig. 1** The experiment. **a** The transverse vibrations of a singly-clamped nanowire are optically read out by focusing a red probe laser beam on its vibrating extremity while recording the reflected intensity fluctuations on a dual photodetector. A counter propagating focused green laser beam generates a tunable optical force field of non-conservative (rotational) nature[16]. **b** Horizontal, $XZ$ maps of the sum/difference DC signals $V_{\oplus/\ominus}(\mathbf{r})$ obtained by piezo-scanning the nanowire in the probe laser waist. The measurement location ⊙ provides quasi-orthogonal projective measurement vectors $\mathbf{e}_{\ominus,\oplus}$ allowing for a 2D-measurement of the nanowire's motion. **c** Steady-state driven trajectories obtained under resonant optical actuation at low optical power (2 µW) for a set of driving frequencies spanning across both eigenfrequencies (300 trajectories (gray curves) from 6670 to 6810 kHz, the red/blue curves are the resonantly driven trajectories). $\mathbf{e}_F$: optical actuation force vector, $\mathbf{e}_{1,2}$: uncoupled eigenvectors. **d** Sketch of the optical force field structure. Measurement positions indicated by ⊗, , feature zero, positive and negative force field rotational respectively. **e** Calibrated thermal noise spectra measured on each measurement channel: $S_{\delta r_{\ominus,\oplus}}[\Omega]$ at position  for increasing green optical power

beam. The 2D optical readout of the NW vibrations is fully described in[17].

The non-conservative force field[16] is generated by an independent counter-propagating tightly focused 532 nm laser beam. The spatial structure of the optical force field experienced by the nanowire is sketched in Fig. 1d and tuned by moving the focusing objective with a $XYZ$ piezo-scanner while its magnitude, proportional to the optical pump power, is varied with an acousto-optic modulator (AOM). The latter can also be simultaneously used to modulate the pump intensity and resonantly drive the NW so as to measure its mechanical response (see Supplementary Note 1) to the local optical force. Three particular positions within the force-applying laser, visible on Fig. 1d, were investigated, each featuring a different local force structure: the central position marked by ⊗ as a zero-rotational test point and the positions  and  for positive and negative force rotational.

**Intrinsic nanowire properties**. Firstly, these were determined in the absence of green light, first by acquiring the noise spectral

densities $S_{\delta r_{\ominus,\oplus}}[\Omega]$ measured on each projected displacement channel on two spectrum analyzers, see the foreground of Fig. 1e. Both fundamental eigenmodes are visible, oscillating around 6.7 kHz with a quality factor of ≈3000 in vacuum ($10^{-3}$ mbar), identical for both modes within 5%. They present a quasi-degenerate character (0.3%) with a frequency splitting of $(\Omega_2 - \Omega_1)/2\pi \approx 20$ Hz, which renders the NW extremely sensitive to the shear components of any external force field[17]. The intrinsic eigenmodes' orientations ($\mathbf{e}_{1,2}$) are determined from the comparison of the peak spectral densities measured on each measurement channel. The displacement associated with each mode is rectilinear along directions forming angles of respectively $\theta_1 = -28°$ and $\theta_2 = +62°$ with the $\mathbf{e_x}$ axis (≈2° uncertainty determined from statistical analysis) so that the modes are found to be perfectly perpendicular in absence of external force field. The impact of the probe beam on the NW dynamics was minimized by reducing optical force field gradients (operation with an enlarged optical waist at low power 15 µW on the optical axis of the probe beam, where any possible residual force field is irrotational). The effective mass of both modes amounts to $M_{eff} = 1.5 \times 10^{-15}$ kg,

corresponding to a spring constant of $\approx 3\ \mu N\ m^{-1}$, which conveys a force sensitivity of $\approx 10\ aN\ Hz^{-1/2}$ at room temperature.

Secondly, response measurements were realized by injecting a small amount of green light (2 μW), which was intensity modulated with the AOM to exert a time-modulated force. After a transitory period the monochromatic drive generates a steady-state trajectory in space, whose projections along $\mathbf{e}_{\ominus,\oplus}$ are measured by the dual 2D readout on a network analyzer featuring two synchronized measurement channels. Sweeping the excitation tone across both eigenfrequencies permits to build the set of steady-state driven trajectories in 2D shown in Fig. 1c after a geometrical reconstruction which is explained in the Supplementary Note 5. On the green optical axis, the optical actuation force vector is quasi-aligned with $\mathbf{e_z}$, see Fig. 1c, and can thus drive both eigenmodes. This representation offers a very straightforward determination of the eigenmodes orientations in space as those correspond to directions where maximal oscillation amplitudes are observed. The results of this method are in excellent agreement with the above measurements derived from 2D thermal noise analysis. The linearity of the response was verified up to large driven amplitudes, approaching 300 nm. Beyond this point the NW may exit the linear measurement area and a mechanical bistability appears at even larger drive amplitudes.

**Observation of eigenmodes warping**. We then insert the NW in a non-conservative force field by increasing the green optical power and positioning the NW on the left side of the green optical waist where the force field rotational is maximal[16] (this position is indicated by  in Fig. 1d). We now focus on the vectorial aspect of the nanowire motion perturbed by rotational forces. We insist that this work deals with rotational force fields that do not entail a dynamical bifurcation as in[16]. To do so we keep the optical powers below the onset threshold or work with optical force field gradient structures $\{g_{ij}\}$ that will not cause a bifurcation at any optical power. Thermal noise spectra (Figs. 1e and 2b) and driven trajectories (Fig. 2a) were recorded for increasing green optical powers, up to 80 μW but not above to avoid the dynamical instability appearing beyond the bifurcation[16]. Strikingly visible on the 2D representations of the driven trajectories (Fig. 2a) appears a strong warping of the eigenmodes orientations which progressively rotate towards a common orientation pointing here around +40° from $\mathbf{e_x}$. The corresponding thermal noise spectra are shown in Figs. 1e and 2b. They show a merging of the eigenfrequencies, a global noise increase and anomalous spectral lineshapes. Eigenfrequencies $\Omega_\pm/2\pi$ and eigenmodes orientations $\theta_\pm$ deduced from both response and noise analyses are reported on Fig. 2c, d. Figure 2f shows the displacement noise spectral densities $S_{\delta r_{\theta_\pm}}[\Omega_\pm]$ at the dressed eigenfrequencies computed for a projective measurement angle aligned with each eigenmode[17]. A noise excess 30 times larger than the undressed thermal noise is observed when approaching the bifurcation. This is larger than the factor 2 expected for the independent summation of two uncorrelated noise spectral densities of similar eigenmodes whose directions become collinear. Radiation pressure noise is largely negligible here and cannot account for this noise excess. A significant increase of the driven response is also visible in Fig. 2a despite the constant drive strength employed (see Supplementary Note 1) and the quasi-perpendicular orientation of the driving vector ($\mathbf{e_F}$) with respect to the dressed eigenmodes orientations ($\mathbf{e_\pm}$) close to the bifurcation. A model is now presented that accounts for all these observations and gives an insight into the physics at play in 2D oscillators cross-coupled through non-conservative (rotational) force fields.

**Model**. The dynamics of the NW deflection $\delta\mathbf{r}(t)$ around its rest position $\mathbf{r_0}$, restricted to the 2 fundamental eigenmodes polarizations follows: $\delta\ddot{\mathbf{r}} = -\overline{\Omega}^2 \cdot \delta\mathbf{r} - \Gamma\delta\dot{\mathbf{r}} + (\mathbf{F}(\mathbf{r_0}+\delta\mathbf{r}) + \delta\mathbf{F} + \delta\mathbf{F}_{th})/M_{eff}$. $\overline{\Omega}^2 \equiv \begin{pmatrix} \Omega_1^2 & 0 \\ 0 & \Omega_2^2 \end{pmatrix}$ is the intrinsic restoring force matrix expressed in the unperturbed $\mathbf{e}_{1,2}$ basis, $\Gamma$ the mechanical damping rate, $M_{eff}$ the effective mass, $\delta\mathbf{F}$ an external probe force and $\delta\mathbf{F}_{th}$ represents the Langevin force vector which drives the NW along each uncoupled axis independently[48] with a white force noise of spectral density $S_F^{th} = 2M_{eff}\Gamma k_B T$. Response measurements confirm the absence of delay on mechanical time scales in the establishment of the optical force experienced by the NW consecutive to an intensity change, as expected for radiation pressure forces[16]. As a consequence, the external force experienced by the NW extremity $\mathbf{F}(\mathbf{r_0}+\delta\mathbf{r})$ only depends on its position within the force field. For small position fluctuations with respect to the characteristic length scale of the force field structure, the force can be linearized as $\mathbf{F}(\mathbf{r_0}) + (\delta\mathbf{r}\cdot\boldsymbol{\nabla})\mathbf{F}|_{\mathbf{r_0}}$. The static force causes a static NW deflection which can amount to $\approx 100$ nm for the largest power employed and redefines the working position within the optical force field. The second linear term–proportional to the NW deflection–represents an additional restoring force which modifies the oscillator's dynamics. The eigenmodes coupling matrix is thus built out of the 4 components of the 2D external force field gradients: $g_{ij}(\mathbf{r_0}) \equiv \frac{1}{M_{eff}}\partial_i F_j|_{\mathbf{r_0}}$. Shear components ($i \neq j$) control the cross-coupling between eigenmodes. In Fourier space, we have $\delta\mathbf{r}[\Omega] = \overline{\chi}[\Omega]\cdot\delta\mathbf{F}_{th}[\Omega]$ where $\overline{\chi}[\Omega]$ is the modified mechanical susceptibility matrix:

$$\overline{\chi}^{-1}[\Omega] \equiv M_{eff}\begin{pmatrix} \tilde{\chi}_{11}^{-1}[\Omega] - g_{11} & -g_{21} \\ -g_{12} & \tilde{\chi}_{22}^{-1}[\Omega] - g_{22} \end{pmatrix}. \quad (1)$$

$M_{eff}^{-1}\tilde{\chi}_{11,22}[\Omega] \equiv M_{eff}^{-1}/(\Omega_{1,2}^2 - \Omega^2 - i\Omega\Gamma)$ are the original uniaxial mechanical susceptibilities. Diagonalizing the susceptibility matrix gives the new eigenmodes labeled with $\pm$ indices whose eigenfrequencies read: $\Omega_\pm^2 \equiv \frac{\Omega_{1\|}^2+\Omega_{2\|}^2}{2} \pm \frac{1}{2}\sqrt{\left(\Omega_{2\|}^2 - \Omega_{1\|}^2\right)^2 + 4g_{12}g_{21}}$ and whose unitary eigenvectors written in the original eigenvectors basis are $\mathbf{e_-} \equiv \frac{1}{\sqrt{g_{12}^2+\Delta\Omega_\perp^2}}\begin{pmatrix} \Delta\Omega_\perp^2 \\ g_{12} \end{pmatrix}$ and $\mathbf{e_+} \equiv \frac{1}{\sqrt{g_{21}^2+\Delta\Omega_\perp^2}}\begin{pmatrix} -g_{21} \\ \nabla\Omega_\perp^2 \end{pmatrix}$. We used $\Omega_{i\|}^2 \equiv \Omega_i^2 - g_{ii}$ and $\Delta\Omega_\perp^2 \equiv \Omega_{2\|}^2 - \Omega_-^2 = \Omega_+^2 - \Omega_{1\|}^2$. The eigenvalues of the susceptibility matrix are defined by $\chi_{\pm\pm}^{-1}[\Omega] \equiv M_{eff}(\Omega_\pm^2 - \Omega^2 - i\Omega\Gamma)$. At first order shear components are responsible for the eigenmode rotation. The scalar product of the eigenvectors follows $\mathbf{e_-}\cdot\mathbf{e_+} \propto \mathbf{rot}(\mathbf{F})\cdot\mathbf{e_y} \propto g_{12} - g_{21}$. In a conservative force field, $g_{12} = g_{21}$, both eigenmodes are equally rotated so that they preserve their original orthogonality as verified in[17]. Instead, the eigenmodes orthogonality is broken in non-conservative force fields, that is, in the case of a real but non-symmetric coupling matrix, as experimentally observed and shown in Fig. 2d.

From these expressions, see Supplementary Note 3, one can compute the projected thermal noise spectra $S_{\delta r_\beta}[\Omega]$ and the steady-state trajectories $\delta\mathbf{r}(t) = \text{Re}(\overline{\chi}[\Omega]\delta F\mathbf{e_F}e^{-i\Omega t})$ driven by a time-modulated force vector $\delta\mathbf{F}(t) = \text{Re}(\delta F\mathbf{e_F}e^{-i\Omega t})$. As in standard cavity optomechanics, our model makes use of a mechanical susceptibility modified by the light field (which can become non-symmetrical when the coupling force presents a non-zero rotational, leading to non-reciprocal coupling) while the Langevin forces and intrinsic damping rates remain unchanged.

**Anomalous thermal noise spectra**. Well visible in Fig. 2b is the distortion of the measured thermal noise spectra into non-

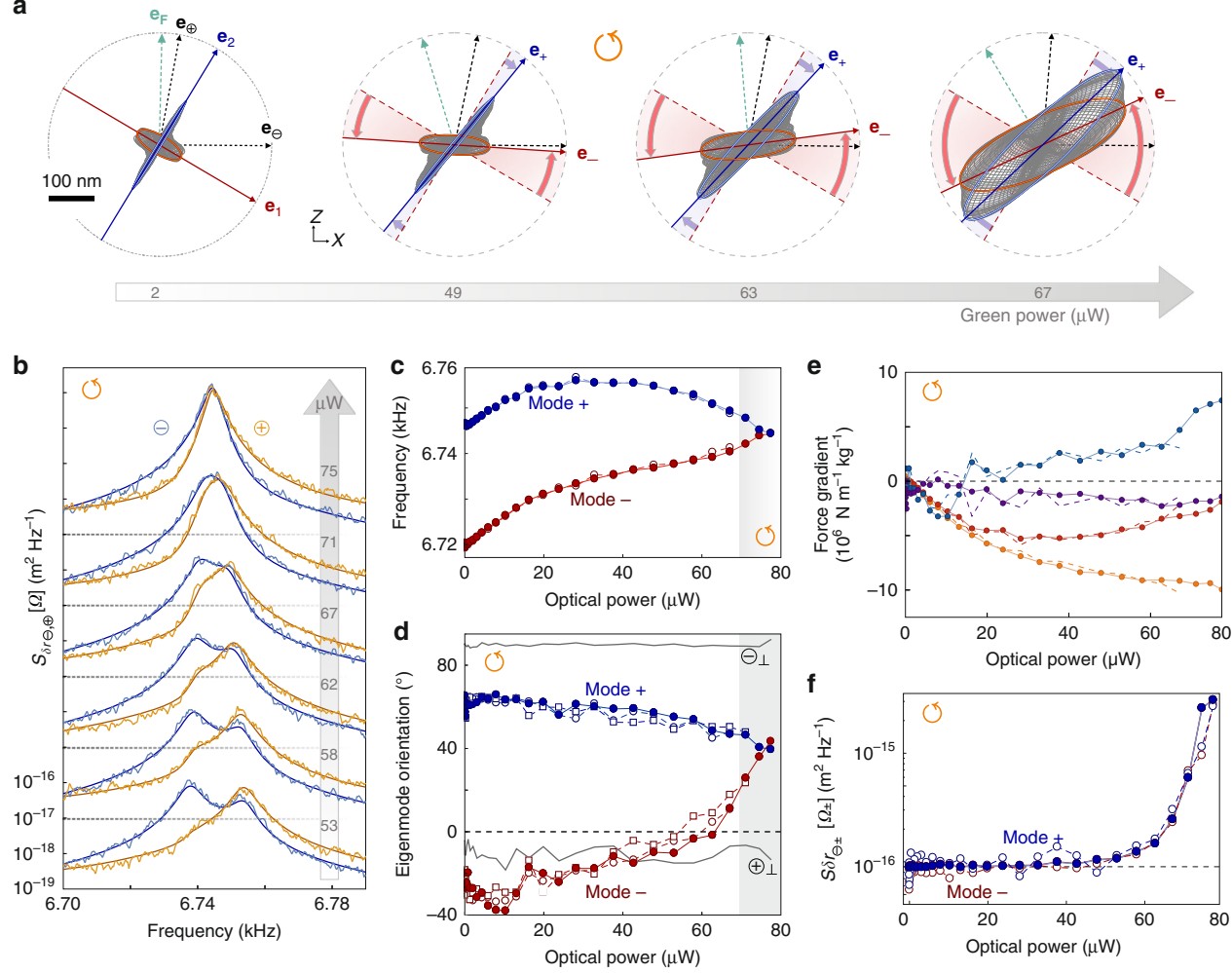

**Fig. 2** Eigenmodes warping. **a** Driven trajectories in the *XZ* plane (gray curves) for increasing static green power but constant driving amplitude. The resonantly driven trajectories at $\Omega_{+/-}$ are highlighted in blue/red. Eigenmodes orientations $\mathbf{e}_\pm$ lose their initial orthogonality and converge towards each other. Also shown are the measurement vectors $\mathbf{e}_{\ominus,\oplus}$ and the drive orientation ($\mathbf{e_F}$), while the eigenmodes rotation is highlighted by shaded red/blue areas. **b** Thermal noise spectra (solid light lines) measured for increasing green power (with a 2-decades vertical offset between each set). Solid dark lines are fits using the equation of $S_{\delta r_{\ominus,\oplus}}[\Omega]$ derived from the model. **c**, **d** Evolution of eigenfrequencies $\Omega_\pm/2\pi$ and eigenmodes orientations $\theta_\pm$ with the green optical power. Angles derived from resonant analysis[17] of thermal noise spectra/driven trajectories are marked as open circles/squares. Filled symbols are deduced from the force field gradients used to fit the thermal noise spectra of panel **b**, which allow a separate determination of the dressed eigenfrequencies and eigenmode orientations. **e** Force field gradients ($g_{11}$ orange, $g_{22}$ red, $g_{12}$ blue, and $g_{21}$ violet) deduced from thermal noise spectral analysis (full circles) and from the determined eigenfrequencies shifts and eigenmodes rotations shown (**c**, **d**) (solid lines) using the inversion relation given in the Supplementary Note 4. **f** Noise spectral densities $S_{\delta r_{\theta_\pm}}[\Omega_\pm]$ deduced from thermal noise analysis (open circles) and from the model Eq. (2) (filled circles), using the fitted force field gradients (**e**)

standard lineshapes presenting asymmetric peaks at large power. Such signatures are absent in conservative force fields[17] and illustrate a large deviation from the normal mode expansion: thermal noise spectra cannot be described anymore as the sum of two spectra featuring Lorentzian mechanical susceptibilities driven by independent Langevin forces. Such a deviation may in some systems originate from the conservative coupling of two resonators with heterogeneous damping[49, 50], but it is not the case here since no damping modification is measured at any optical power. Instead, each couple of thermal noise spectra can be very well simultaneously fitted with the model (solid lines in Fig. 2b) featuring unmodified identical damping rates. The unperturbed mechanical properties ($\Omega_{1,2}$, $\Gamma$, $\theta_{1,2}$, and $M_{\rm eff}$) were fixed to the previously determined values, so that the only fitting parameters are the 4 components of the force field gradients $g_{ij}$ which are reported in Fig. 2e. They vary with the green optical power but deviate from the expected linearity at large optical powers. This

deviation is due to the static deflection which displaces the NW in the force gradient landscape. The measured shear gradients $g_{12}$ and $g_{21}$ are different, which directly demonstrates the non-conservative nature of the optical force field created by the green laser at the measurement position.

The above determination of the force field gradients results from fits of the entire NW thermal noise spectra in 2D. Another method makes use of the measured perturbation of the eigenmodes (rotations and frequency shifts from Fig. 2c, d) following the protocol exposed in[17] and Supplementary Notes 4 and 7. The $g_{ij}$ obtained from this method are also reported in Fig. 2e (solid lines) and a good agreement is observed between both methods. The derivation based on spectral analysis still operates close to the bifurcation while it becomes difficult to determine the eigenmode properties when they are not spectrally resolved anymore (gray shaded area in Fig. 2c, d).

**Excess of thermal noise.** Figure 2f shows the evolution with increasing pump power of $S_{\delta r_{\theta_\pm}}[\Omega_\pm]$, the thermal noise measured at resonance in the optimum readout configuration, that is, for a measurement vector aligned with the eigenmode. The model predicts (see Supplementary Note 3) a noise spectral density of:

$$S_{\delta r_{\theta_\pm}}[\Omega] = \frac{2\Gamma k_B T}{M_{eff}\left((\Omega_\pm^2 - \Omega^2)^2 + \Omega^2\Gamma^2\right)}\left(1 + \frac{(g_{21} - g_{12})^2}{\left(\Omega_\mp^2 - \Omega^2\right)^2 + \Gamma^2\Omega^2}\right).$$

(2)

The first fraction directly corresponds to $\frac{2k_B T}{\Omega}\mathrm{Im}\chi_{\pm\pm}[\Omega]$, which is the thermal noise expected assuming the validity of the FDR[48, 51] when measuring along an eigenmode direction $\mathbf{e}_\pm$. As a consequence, the fraction in the parenthesis represents the expected excess of noise, which is governed by $(\mathrm{rot}\,\mathbf{F})^2$. As observed experimentally, see Fig. 2b, the noise excess is not homogeneously spectrally distributed, but peaked at the second eigenmode frequency. As such one can conclude that it is not possible to define a standard effective temperature[48, 52] in our system. The 30-fold increase of the resonant noise spectral densities is well fitted by the above expression evaluated at $\Omega_\pm$ using the experimentally derived force field gradients, see Fig. 2f. Since a single 1D measurement cannot discriminate a noise increase from an eigenmode rotation, because they both cause a change of the measured signal strength (thermal noise or response), we underline that noise thermometry of 2D coupled systems can be subject to large misinterpretations if a proper 2D readout is not implemented.

**Deviation from the fluctuation-dissipation relation.** The strong asymmetry observed in the thermal noise spectra, the obvious deviation from the normal mode expansion and the measured excess of noise naturally call for an investigation of the FDR in our system. For a generic oscillator at thermal equilibrium in the linear response regime, the FDR connects the thermal noise spectrum $S_{\delta r_\beta}[\Omega]$ measured in an arbitrary direction $\mathbf{e}_\beta$ to the imaginary part of the NW mechanical susceptibility[48, 51, 53] according to:

$$S_{\delta r_\beta}^{FDR}[\Omega] = \frac{2k_B T}{|\Omega|}\left|\mathrm{Im}\chi_{\beta\beta}[\Omega]\right|,$$

(3)

where we have used the tensorial form of the 2D mechanical susceptibility: $\chi_{\mu\nu}[\Omega] \equiv \mathbf{e}_\mu \cdot \overline{\chi}[\Omega] \cdot \mathbf{e}_\nu$ corresponding to a drive along $\mathbf{e}_\nu$ and a measurement along $\mathbf{e}_\mu$. The mechanical susceptibility involved in the FDR corresponds to the one determined from a response measurement realized with a test force of the same spatial profile as the readout mode[48] and oriented along the measurement vector $\mathbf{e}_\beta$.

The comparison between the noise spectra given by Eqs. (2) and (3) evaluated for $\mathbf{e}_\beta = \mathbf{e}_\pm$ shows that the fluctuations in our system are not expected to verify the FDR. We will now directly experimentally verify this conjecture, through a simultaneous measurement of both the thermal noise spectra and the 2D mechanical susceptibility.

Measurements were first performed on the left side of the optical axis (position in Fig. 1d), see Fig. 3a–c, using 63 μW of pump power which is sufficient to break the eigenmode orthogonality while remaining spectrally resolved (see Fig. 2d). The orientation and magnitude of the driving force vector $\delta F\mathbf{e}_F$ are determined as fit parameters of the response measurements

(Fig. 3b) with the mechanical susceptibility of Eq. (1) using the force field gradients $g_{ij}$ deduced from the fit of the 2D Brownian motion (Fig. 3c). The results for the two measurement channels (both in amplitude and phase) are shown in Fig. 3b and accurately catch the data on a very large dynamical range (>40 dB) for a driving force of 270 aN oriented at 94° from $\mathbf{e}_x$. In this situation, the driving vector is quasi aligned with the $\mathbf{e}_\oplus$ readout vector, see Fig. 3a. The axial susceptibilities $\chi_{\ominus\ominus}$ and $\chi_{\oplus\oplus}$ can then be evaluated using expression (1) and the set of determined $g_{ij}$. Note that a weak coherent parasitic background was added to fit the responses to account for parasitic electrical crosstalk. The thermal noise spectra expected from the expression for the fluctuation dissipation relation (3) and computed from $\chi_{\ominus\ominus}$ and $\chi_{\oplus\oplus}$ were then reported in Fig. 3c (solid dark lines) and compared to the spectra measured on each measurement channel. While the two quantities coincide far from resonance, a strong deviation is observed in the vicinity of eigenfrequencies, larger than 10 dB on each channel. As stated above, the fact that the deviation is maximal at resonance is well captured by Eq. (2). This discrepancy demonstrates the deviation from the FDR in our system.

A similar control measurement set was realized in absence of non-conservative force field in position $\otimes$ using the lowest static green optical power (2 μW) which still experimentally allowed identical intensity modulation amplitude to maintain the same driving force, see Fig. 3d–f. Here also, the only free parameters were the orientation and strength of the driving force vector used to probe the mechanical responses. In this case an excellent agreement is obtained between the two members of Eq. (3) which validates the procedure. Without surprise, in absence of non-conservative force field when the eigenmodes orthogonality is preserved, the FDR is verified in our system.

Finally, measurements were performed on the other side of the beam at position in Fig. 1d in an opposite rotational where eigenmodes also loose their orthogonality but eigenfrequencies are repelled from each other instead of merging. The analog of Fig. 2 at that position is given in the Supplementary Note 2. A large deviation from the FDR is similarly observed at large pump power (80 μW), see Fig. 3i, especially pronounced at the eigenmodes resonances. This underlines that the deviation is not due to the frequency merging mechanism but instead—as will now be discussed—to the eigenmodes orthogonality breaking induced by the non-conservative force field.

The rotational force field is indeed responsible for bringing the system in an out-of-equilibrium steady state, violating a fundamental—but often delicate to verify—hypothesis required to establish the FDR.

**Non-axial contributions.** Inspecting the geometrical properties of the tensorial susceptibility permits a phenomenological understanding of the deviation. For the purpose of illustration, we focus on the situation where the measurement vector is aligned with one eigenmode: $\mathbf{e}_\beta = \mathbf{e}_+$. A drive vector aligned with the eigenmode $\mathbf{e}_F = \mathbf{e}_+$ generates a driven displacement confined along the eigenmode orientation characterized by the uniaxial component of the tensorial susceptibility $\chi_{++} = 1/M_{eff}(\Omega_+^2 - \Omega^2 - i\Omega\Gamma)$. This is the axial susceptibility involved in the FDR expression (3) when measuring along $\mathbf{e}_+$. Its resonance frequency differs from the uncoupled one but it does so much too weakly (<0.5% relative frequency shift) to account for the observed excess of noise whose origin is now sought in the non-axial susceptibility terms, that is, the component that characterizes the response in one direction to a force exerted along another direction.

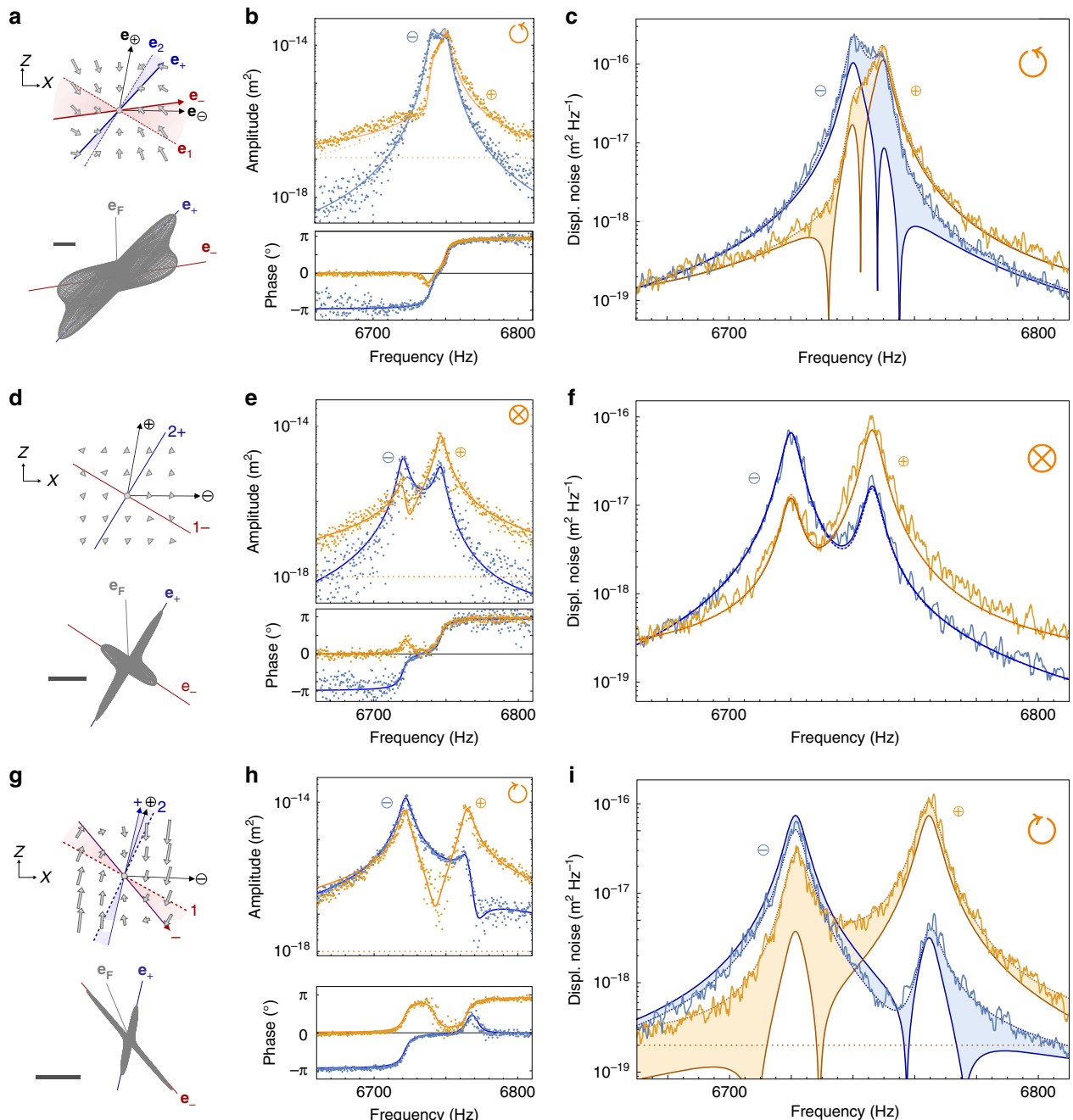

**Fig. 3** Deviation from the fluctuation-dissipation relation. **a** Spatial structure of the optical force field at built from the force gradients deduced from thermal noise analysis for 63 μW of pump power (100 nm scalebar). The 2D representations of the driven trajectories are built from the response measurements–amplitude and phase–on each readout channel (⊖, ⊕) obtained from one single frequency sweep and shown as dots in **b**. **c** Corresponding thermal noise spectra (solid light lines) measured on each channel, jointly fitted (dashed lines) to estimate the $g_{ij}$ parameters. The values of the latter were then fixed to fit the response measurements (solid lines in **b**) with the actuation amplitude (found to amount to 270 aN) and direction (represented as $\mathbf{e_F}$ in **a**) as only free parameters. This fitting step completed the determination of the susceptibility matrix whose particular uniaxial components $\chi_{\ominus\ominus}$ and $\chi_{\oplus\oplus}$ were injected in the second member of Eq. (3) to compute the thermal noise spectra expected from the FDR (solid dark lines on **c**). Pronounced discrepancies (>10 dB) between these estimates derived from the fluctuation dissipation relation and the direct experimental traces are emphasized as colored areas. Panels **d**–**f** are taken in position ⊗ at low optical power (2 μW) where the shear force field gradients can be neglected. There the measured spectra are in excellent agreement with the prediction of the fluctuation dissipation relation. Panels **g**–**i** are taken on the other side of the optical axis ( ) with an opposite rotational where no frequency merging occurs, see Supplementary Note 2. A similar substantial deviation is observed

Strikingly, as shown in Fig. 4a (gray curves), a drive vector almost perpendicular to $\mathbf{e_-}$ can generate a significant displacement along $\mathbf{e_+}$, larger than the one obtained with a drive vector aligned with $\mathbf{e_+}$ (blue curve). Indeed, the maximum resonantly driven displacement along one eigenmode orientation is shown to

be obtained for a drive applied perpendicularly to the other eigenmode: $\theta_F^{\pm\,\mathrm{opt}} = \theta_{\mp} + \pi/2$ (see Supplementary Note 5). In absence of perpendicularity breaking, one recovers the intuitive result that the drive of one eigenmode is most efficient if it is applied in the direction of this eigenmode. The counter-intuitive

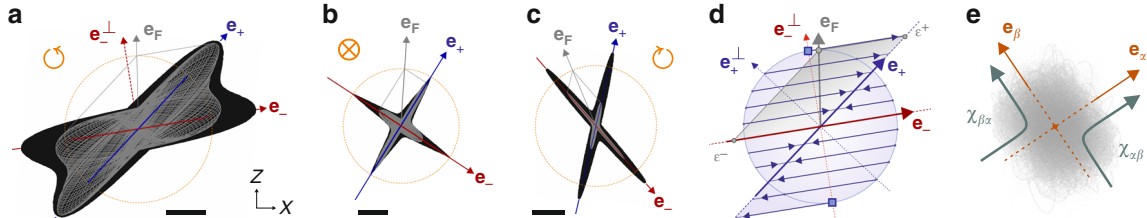

**Fig. 4** Tensorial structure of the 2D mechanical response. **a–c** Driven trajectories obtained at the three different locations of Fig. 3 for varying orientations of the drive vector (fixed magnitude of 270 aN). Red/blue curves are expected when driving along $\mathbf{e}_{-/+}$, reaching a maximum amplitude represented as a dashed orange circle. Gray curves are measured for the drive orientation of Fig. 3. Black curves represent the entire set of trajectories obtained for all drive orientations, see Supplementary Note 5. (100 nm scalebars) **d** Gray: Illustration (in the situation) of the expansion of a unitary vector $\mathbf{e_F}$ on the non-orthogonal $\{\mathbf{e}_{\pm}\}$ base: $\mathbf{e_F} = \varepsilon^{-}\,\mathbf{e}_{-} + \varepsilon^{+}\,\mathbf{e}_{+}$. Its contravariant coordinate $\varepsilon^{+}$ is maximized when the force vector is aligned with the covariant element $\mathbf{e}_{-}^{\perp}$ of the dual base, perpendicular to $\mathbf{e}_{-}$ (angle marked by a blue square). **e** Sketch of a possible couple of transverse mechanical susceptibilities whose asymmetry determines the magnitude of the deviation from the FDR according to Eq. (4)

character of this observation arises from the fact that one expects to optimize the drive efficiency of an eigenmode $\mathbf{e}_{+}$ by maximizing the driving force coordinate along this direction, but often forgets that the coordinate to maximize is the contravariant coordinate $\varepsilon^{+}$ defined by: $\mathbf{e_F} = \varepsilon^{-}\,\mathbf{e}_{-} + \varepsilon^{+}\,\mathbf{e}_{+}$ rather than the covariant coordinate $\varepsilon_{+} = \mathbf{e_F} \cdot \mathbf{e}_{+}$. Indeed the two coincide in more familiar orthogonal bases (see orthogonal projections in the unperturbed situation of Fig. 4b) but differ in warped bases (see Fig. 4a, c). The geometric representation given in Fig. 4d can be used to determine the contravariant coordinate's expression: $\varepsilon^{+} = \mathbf{e_F} \cdot \mathbf{e}_{-}^{\perp}/\sin(\arccos(\mathbf{e}_{-} \cdot \mathbf{e}_{+}))$. The maximum enhancement of the non-axial susceptibility is therefore given by $\varepsilon_{\max}^{+} = (1 - (\mathbf{e}_{-} \cdot \mathbf{e}_{+})^2)^{-1/2}$, which establishes a direct connection between the increased mechanical response and the eigenmodes orthogonality breaking. As a consequence, in a non-conservative force field, the transverse components of the tensorial susceptibility ($\chi_i^{\perp} \equiv \mathbf{e}_i \cdot \chi \cdot \mathbf{e}_i^{\perp}$) allow each mode to be simultaneously driven by the second, lateral uncorrelated Langevin force. This mechanism quantitatively accounts for the observed excess of noise analyzed in Fig. 2f.

**Modified fluctuation dissipation relation.** Further developing the model exposed above, it is possible to demonstrate, see Supplementary Note 6, that the deviation from FDR observed in such a non-reciprocally coupled system verifies

$$\sum_{\mu=\beta,\beta^{\perp}}\left(S_{\delta r_{\mu}} - \frac{2k_{\mathrm{B}}T}{|\Omega|}\left|\mathrm{Im}\chi_{\mu\mu}\right|\right) = S_{\mathrm{F}}^{\mathrm{th}}\left|\chi_{\alpha\beta} - \chi_{\beta\alpha}\right|^2, \qquad (4)$$

where $\{\mathbf{e}_{\alpha}, \mathbf{e}_{\beta}\}$ represents an arbitrary couple of two orthogonal orientations, as sketched in Fig. 4e. This expression represents a patch for the FDR in our system, valid in the out-of-equilibrium but stationary regime explored here. It underlines the role of non-symmetric coupling ($\chi_{12} - \chi_{21} \propto g_{12} - g_{21}$) in the observed deviation and the importance to determine the entire 2D tensorial mechanical susceptibility to describe the fluctuations of a multi-mode nanomechanical system. We note that the quadratic dependence of the deviation in rot$\mathbf{F}$ originates from the fact that the NW is exploring a rotational force field along Brownian trajectories whose statistics is unbalanced by the non-conservative force (see Supplementary Note 8). Equation 4 also suggests a very generic methodology to check whether the FDR applies in such a system, that consists in measuring the resemblance between any couple of opposite transverse susceptibilities $\chi_{\alpha\beta}$, $\chi_{\beta\alpha}$. This approach therefore only requires the analysis of response measurements that can be repeated and averaged and realized with a small resolution bandwidth in order not to be limited by the

measurement background noise. This remark might be of great use for systems where the experimental verification of the FDR is rendered delicate by a low signal-to-noise ratio that would not allow for the measurement of thermal motion in real time. This situation is often encountered in optomechanical systems at low phonon occupancy where it becomes difficult to verify the equilibrium hypothesis.

The transverse response asymmetry can be clearly visualized in the configuration of Fig. 4a with $\{\mathbf{e}_{\alpha}, \mathbf{e}_{\beta}\} = \{\mathbf{e}_{-}, \mathbf{e}_{-}^{\perp}\}$. For a drive along $\mathbf{e}_{-}$, no transverse motion is generated so that $\mathbf{e}_{-}^{\perp} \cdot \chi \cdot \mathbf{e}_{-} = 0$. On the contrary, driving along $\mathbf{e}_{-}^{\perp} \approx \mathbf{e_F}$, produces a significant displacement along $\mathbf{e}_{-}$ (gray curves) so that $\mathbf{e}_{-} \cdot \chi \cdot \mathbf{e}_{-}^{\perp} \neq 0$.

The weighted spectral integral of Eq. (4) yields the power injected by the non-conservative force field and fully dissipated by damping forces (see Supplementary Note 8), according to the Harada-Sasa theorem[54, 55] for any out-of-equilibrium systems. In contrast with the original theorem, we propose here a spectral, non-integrated formulation of the deviation to the FDR applied to non-reciprocally coupled systems. The injected power also corresponds to the entropy production rate in the system[56], see Supplementary Note 8, which further underlines the key role of the non-axial susceptibilities from a thermodynamical point of view. The imbalance between opposite transverse susceptibilities appears in warped, non-orthognal eigenbases since $\chi_{\alpha\beta} - \chi_{\beta\alpha} \propto \mathbf{e}_{+} \cdot \mathbf{e}_{-}$: the last equation therefore draws a direct link between the breaking of the eigenmodes orthogonality and the violation of the fluctuation-dissipation relation in non-reciprocally coupled multimode systems. Hence, this equation is important for the description of fluctuations in such out-of-equilibrium systems and represents an important result of this work.

In summary, we reported on eigenmodes warping, distortion of thermal noise spectra and deviation from the FDR when inserting a multimode 2D nanoresonator in a non-conservative but non-viscous coupling force field[35]. Our model quantitatively accounts for all observations and accurately describes the fluctuations and driven dynamics of the strongly and non-reciprocally coupled 2D system. This work experimentally validates the principle of force field sensing[17] based on the recording of eigenmodes orientations and frequency shifts in the yet unexplored non-conservative case. It sheds light on some subtleties arising in noise thermometry in strongly confined optical fields and points out the importance of the transverse mechanical susceptibilities in the observed phenomenology. Similar signatures should be observable in any non-reciprocally coupled dual physical systems and in particular in multimode cavity-optomechanics. This system may also be of great interest to test new formulations of

fluctuation theorems[57–60] based on 2D trajectory analysis. It is also an interesting platform for investigating real time dynamics in rotational force fields[35].

**Data availability**. The data that support the findings of this study are available from the corresponding author upon request.

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

## Acknowledgements

We warmly thank the PNEC group at ILM, C. Elouard, A. Gloppe, F. Fogliano, L. Mougel, J. P. Poizat, G. Bachelier, J. Jarreau, C. Hoarau, E. Eyraud and D. Lepoittevin. This project is supported by the Agence National de la Recherche (FOCUS), the European Research Council (StG-2012-HQ-NOM and POC-2017-CARTOFF projects) and the LANEF framework (ANR-10-LABX-51-01, project CryOptics).

## Author contributions

All authors contributed to all aspects of the work.

## Additional information

**Competing interests:** The authors declare no competing interests.

