## [Peer Review File · Nature Communications]

Reviewers' comments:

Reviewer #1 (Remarks to the Author):

This article describes a very interesting experiment on the noise of a nano-wire which may be employed to measure very weak forces. The presented analysis is necessary in order to safely use this device in practical applications. The main results is that a non-conservative force applied to the system couples the eigenmodes of the nanowire and the thermal fluctuations violate the FDR. The authors justify their results using a model based on coupled harmonic oscillators which perfectly fit the experimental data.

For these reasons there are no doubts that the very original results of this article merit to be published in Nature Communications. I have no particular criticisms on the data and the model, however before publication the following comments on the presentation should be taken into account to improve the quality of the presentation.

The article is very complete and with the help of SI one finds all the needed informations. However it is rather difficult to read because in the main text there are too many informations and the most important points are not clearly pointed out. The figures are overcrowded. Most importantly the calibration protocol of the force, which is described in fig.2 and 3, should be more explicitly described for example in a section of SI, making clear references to the used equations.

Reviewer #2 (Remarks to the Author):

The work describes experiments on the vibrations of a suspended carbon nanotube. The nanotube has two modes, which are called longitudinal, even though Fig. 1 makes an impression that the transverse displacement of the nanowire is measured. The figure shows eigenfrequency of ~ 6.7 kHz, whereas the figure caption indicates frequencies in the range of 6700 kHz, which seems too high. The paper first reports the linear response measurements and the measurements of the spectra of thermal fluctuations. The nanowire is then illuminated by a strongly focused intense laser beam. Because the beam is spatially nonuniform, the force from the beam depends on the nanowire position, leading to mode mixing (note that such mixing should arise also from the combination of radiation pressure and mode nonlinearity). For some reason, the force does not linearly scale with the beam intensity.

The major observation of the paper is that the force from the beam is non-conservative. Therefore the vibrations become thermally nonequilibrium. The paper reports that the standard fluctuation-

dissipation relation does not apply. Instead, the power spectrum has an extra contribution from the curl of the force. It is compared with “the experimentally derived force field gradients”, which is confusing in the reference to a curl. The effect is described by the expression for the power spectrum, which is called the “novel modified version of the fluctuation dissipation relation”. In fact, what is proposed is just a solution of a linear problem.

This is a regular paper, I do not see any big discoveries or far-reaching consequences. The experimental observations are nice. The theory is elementary. A major deficiency is that the nature of the nonconservative force is not understood and its dependence on the beam intensity is not described. This is, to me, the most interesting physics question.

The way the paper is written is inconsistent with the significance of the results. I find it annoying that simple things are presented as something very sophisticated. For example, what is the goal of writing a standard expression from the Feynman and Hibbs book, which contains a single integral over time, as a double integral with a delta-function [eqs. (S75) and (S76)]? Incidentally, path integrals are not needed for the presented simple and straightforward calculation in the first place; the analysis can be found in standard texts on Markov processes, like Risken’s or Gardiner’s books; see also references in Ref.29.

The terms used in the paper are nonconventional. For example, what is meant by “nonsymmetric coupling” in the Penning trap?- The mode coupling in this systems has been studied in much detail in the Gabrielse group after the review [26].

Many clauses in the paper are hard to decipher. Just on page 1 you see “is of non-conservative nature as it presents a non-zero rotational.”, or “topologically non-conservative force field”, and this list can be continued. This makes it hard to read the paper and to see what it is about.

I do not recommend publishing the paper in Nature Communications.

Reviewer #3 (Remarks to the Author):

The major claims of the paper are finding eigenmode orthogonality breaking, non-reciprocity in eigenmode cross-coupling, an excess of anomalous displacement noise, and a violation of the fluctuation-dissipation theorem.

I advise publication of this manuscript, albeit with major revisions.

The major concern is the length and the focus of the manuscript. Most of the claims were found in reference 16 (eigenmode non-orthogonality, excess of noise, non-reciprocity). The new result here is the analysis of the thermal spectra when the modes become non-orthogonal. This is given consideration, but the findings before the section "Excess of thermal noise" is repetition from reference 16. The other parts of the paper are important for understanding some of the new results, but that should be handled with referencing their 2014 paper (reference 16). Much of the supplementary information is repetitive of the other paper as well, and should be removed and referenced.

Given the focus of the manuscript, the referee advises a change in title which emphasizes the new result; that is, something that reflects the change in susceptibility for two cross-coupled non-reciprocal modes. 'Anomalous dynamics' is vague. "Anomalous displacement fluctuations" would be more descriptive.

The referee advises removing much of the written material before the analysis of the thermal spectra. The referee also advises moving most of Figure 2 to the supplementary information since it seems to be calibration and not important for the analysis.

The referee also calls attention to the additional points:

1) Although it's a small part of this paper, the referee feels the use of the term "topological bifurcation" is confusing. The fact the nanowire follows an ellipse rather than a line after the bifurcation point is immaterial. There is no effect in the spectral data which reflects the change in topology, other than the limit cycle itself. Higher dimensional nonlinear dynamical systems (which this system is given the fact they have negative dissipation past the bifurcation via optomechanical gain, which must be restrained by nonlinear dissipation) show all kinds of multidimensional trajectories (in physical space). The referee does not find this term in the literature of Rayleigh–Bénard convection, where the fluid forms 'topological' patterns upon bifurcation. Using the word 'topological' is confusing and should be removed. The referee advises classifying the bifurcation in accord with the nonlinear dynamics literature.

2) The referee finds the following statement confusing, "It also suggests a very generic methodology to check whether the FDR applies in such a system, that consists in measuring the resemblance

between any couple of opposite transverse susceptibilities

. This approach therefore only requires the analysis of response measurements that can be repeated and averaged. This remark might be of great use for systems where the experimental verification of the FDR is rendered delicate by a low signal-to-noise ratio that would not allow for the measurement of thermal motion." How so? Aren't the authors claiming that theirs is the only system with these properties? Maybe they could give a reference for an example.

2) The authors use the word 'immerse'. This is a word for dipping in fluids, not insertion into laser fields (they actually turn on the field around the nanowire so that's not right either). The referee advises a more accurate description of the experimental procedure.

3) The authors state "Their ultralow-mass is responsible for a sizeable thermal position noise, that can spread over distances approaching their intrinsic dimensions". The first half of this statement seems incorrect. Isn't position noise proportional to $kT/\text{stiffness}$? Their device has an extreme aspect ratio which would lead to a very low stiffness. Their device actually has a larger mass than typical nanomechanical devices.

4) The figures are very cramped and difficult to read. The referee suggests removing most of Figure 2 and splitting up or expanding figure 3.

Response to referees' comments

Reviewers' comments:

Reviewer #1 (Remarks to the Author):

This article describes a very interesting experiment on the noise of a nano-wire which may be employed to measure very weak forces. The presented analysis is necessary in order to safely use this device in practical applications. The main result is that a non-conservative force applied to the system couples the eigenmodes of the nanowire and the thermal fluctuations violate the FDR. The authors justify their results using a model based on coupled harmonic oscillators which perfectly fit the experimental data.

For these reasons there are no doubts that the very original results of this article merit to be published in Nature Communications. I have no particular criticisms on the data and the model,

however before publication the following comments on the presentation should be taken into account to improve the quality of the presentation.

We thank the referee for pointing out the interest of the work.

The article is very complete and with the help of SI one finds all the needed informations. However it is rather difficult to read because in the main text there are too many informations and the most important points are not clearly pointed out. The figures are overcrowded.

We have modified the figures to make them more readable and have also clarified the manuscript. Please refer to the list of changes below.

Most importantly the calibration protocol of the force, which is described in fig.2 and 3, should be more explicitly described for example in a section of SI, making clear references to the used equations.

As requested, we have added a dedicated section in the SI describing the signal analysis routines employed.

Reviewer #2 (Remarks to the Author):

The work describes experiments on the vibrations of a suspended carbon nanotube. The nanotube has two modes, which are called longitudinal, even though Fig. 1 makes an impression that the transverse displacement of the nanowire is measured. The figure shows

eigenfrequency of ~6.7 kHz, whereas the figure caption indicates frequencies in the range of 6700 kHz, which seems too high.

Even if higher order eigenmodes are accessible, we are working here only with the 2 first eigenmodes, which oscillate transversally at quasi identical frequencies (that are indeed represented schematically on Fig 1a, we inserted a reference in the text to this figure to avoid any further confusion) . We removed the term longitudinal which could be mixed with elongation or breathing modes, and now exclusively call them “ the 2 fundamental transverse eigenmodes”.

They oscillate around 6.7 kHz, thanks for pointing out the error in the punctuation (6,700 kHz vs 6.700 kHz) in one of the figure.

The paper first reports the linear response measurements and the measurements of the spectra of thermal fluctuations. The nanowire is then illuminated by a strongly focused intense laser beam. Because the beam is spatially nonuniform, the force from the beam depends on the nanowire position, leading to mode mixing (note that such mixing should arise also from the combination of radiation pressure and mode nonlinearity).

As mentioned in the article, mechanical non-linearities arise at larger drive amplitudes and do not play any role here.

For some reason, the force does not linearly scale with the beam intensity.

No, at a given position, the force is linear with the optical intensity.

As explained in the manuscript, the deviation observed in Fig. 2 in the derived g_{ij} from a perfectly linear behaviour in the injected power is a consequence of the static contribution of the radiation pressure force: when we increase the optical power, the rest position of the nanowire, around which it oscillates, is displaced in the green force field. Since the latter is non-homogeneous, the force field derivatives do not vary linearly either when the static displacement is too important. This static deflection is thus responsible for a change in the system working point, but the subsequent analysis of its fluctuation/response around one rest position is not impacted.

The major observation of the paper is that the force from the beam is non-conservative. Therefore the vibrations become thermally nonequilibrium. The paper reports that the standard fluctuation-dissipation relation does not apply. Instead, the power spectrum has an extra contribution from the curl of the force. It is compared with “the experimentally derived force field gradients”, which is confusing in the reference to a curl.

the 2D force field presents 4 spatial derivatives, or gradients, ($dxF_x, dxF_z, dzF_x, dzF_z$) which all have an impact on the nanowire dynamics. The curl/rotational of the force field is simply $dxF_z - dzF_x$.

The non-conservative nature of the optical force field means that the force field does not derive from a potential energy (as $F = -\text{grad } U$) which is here a consequence of the fact that it has a non-zero curl/rotational ($\text{rot } F$ different from zero).

The effect is described by the expression for the power spectrum, which is called the “novel modified version of the fluctuation dissipation relation”. In fact, what is proposed is just a solution of a linear problem.

Right, the physics explored here is fully linear, but we believe this does not impair the novelty of our findings.

This is a regular paper, I do not see any big discoveries or far-reaching consequences. The experimental observations are nice. The theory is elementary.

Indeed we do not need a more involved theory to fully explain our results. Despite the “elementary” character of the theory, it brings a lot of novelty to the fields of optomechanics or nanomechanics and propose a novel implementation for out of equilibrium physics

We do not agree on the lack of novelty, we have established a list of 8 points through which the novelty and importance of our work should be considered:

- 1) Eigenmode orthogonality breaking. This was neither reported nor explored elsewhere before.
- 2) Anomalous thermal noise: violation of the normal mode expansion, and observation of an excess of noise by a factor 30.
- 3) Deviation from the FDR. Development of a protocol to properly establish the deviation in 2D multimode systems.
- 4) Theoretical prediction and direct experimental confirmation of the key role of the transverse components of the mechanical susceptibility.
The subtleties arising with non-symmetric susceptibility matrixes were neither found elsewhere, nor explored in our own articles.
- 5) 2D Patch of the fluctuation dissipation relation valid in the out of equilibrium case. While there is a very rich literature on the violation of the FDR, to our best knowledge, our expression was not found in any previous article. The closest expression is the one employed in the Harada-Sasa formulation, but no direct connection is made to the non-reciprocal mechanical susceptibility. It is also one of the first experimental articles which brings advanced out-of-equilibrium physics concepts in the field of optomechanics.
- 6) An experimental strategy is given in order to verify if the system can be found out of equilibrium through a simple measurement of the 2D mechanical susceptibility (determination of χ_{12} - χ_{21}), without prior hypothesis on the external force field. It is essential to underline that the proposed method is the only one available when one does not have access to trajectories of the system in real time (but only to time average spectral informations as often encountered in optomechanics especially at low phonon occupancy).

- 7) We extend the principle of force field sensing exposed in ref. 17 to the non-conservative case. It remains valid despite all of the experimental “surprises” observed in that situation.
- 8) The 2D analysis of driven trajectories proposes a novel complementary method (in addition to thermal noise 2D analysis) to determine the eigenmode orientations. This opens the way to faster force field imaging based on coherently driven trajectories.

A major deficiency is that the nature of the nonconservative force is not understood and its dependence on the beam intensity is not described. This is, to me, the most interesting physics question.

We did not expound upon this point in this article because the rotational force was already explored in our previous work, and is already known to the community of optical trapping.. The novelty here does not consist in the existence of this force (see the references cited in the manuscript, we have added a more recent one: Wu2009 where the rotational force field was measured in an optical tweezer experiment) but on its consequences on Brownian underdamped motion - what was never observed with damped colloidal particles trapped in water - and on a new modified formulation of the fluctuation-dissipation relation valid in rotational force fields (and more broadly in non-symmetrically coupled multidimensional systems.), see the above comments on this work novelties.

The focused green laser beams applies a radiation pressure force which varies with the position within the optical waist. The corresponding spatial force field presents a non-zero rotational (curl) so that the force field is non-conservative in the sense that it does not derive from a potential energy.

The force scales linearly with the optical power. For a given green laser intensity, the static deflection is fixed. If one then realizes response measurements by modulating the green power (fixed DC + modulated AC) and sweeping the drive frequency across the 2 eigenmodes, then the response is fully linear in the modulation strength.

As explained above and in the article, the fact that the force field gradients do not evolve linearly with the optical power is a consequence of the static deflection which displaces the rest position of the nanowire. However, as verified experimentally, the response of the system remains fully linear in all of the position explored for the drive amplitudes employed.

The way the paper is written is inconsistent with the significance of the results. I find it annoying that simple things are presented as something very sophisticated. For example, what is the goal of writing a standard expression from the Feynman and Hibbs book, which contains a single integral over time, as a double integral with a delta-function [eqs. (S75) and (S76)]? Incidentally, path integrals are not needed for the presented simple and straightforward calculation in the first place; the analysis can be found in standard texts on Markov processes, like Risken's or Gardiner's books; see also references in Ref.29.

We have not found our theoretical derivations elsewhere, in the sense that we work here on an underdamped 2 mode resonator (in contrast to optical tweezers of ref 29 and elsewhere). The thermodynamical equations in the supplementary material are inspired as stated from the derivation of Zamponi et al, so we have followed exactly their approach and formalism.

The terms used in the paper are nonconventional. For example, what is meant by “nonsymmetric coupling” in the Penning trap? - The mode coupling in this systems has been studied in much detail in the Gabrielse group after the review [26].

The Lorentz force ($q \mathbf{v} \times \mathbf{B}$) is non-conservative in the sense that it does not derive from a potential energy. This term couples the motions perpendicular to the magnetic field but with an opposition sign: $dv_x/dt \propto v_z$ and $dv_z/dt \propto v_x$.

We note that this coupling mechanism is not instantaneous as in our case (force proportional to the nanowire deflection) but delayed (proportional to the speed).

However we agree that this formulation was not straightforward and chose to remove the reference to this example of another non-conservative force from the article.

Many clauses in the paper are hard to decipher. Just on page 1 you see “is of non-conservative nature as it presents a non-zero rotational.”, or “topologically non-conservative force field”, and this list can be continued. This makes it hard to read the paper and to see what it is about.

We have tried to improve the manuscript readability. We modified "is of non-conservative nature" into "is non-conservative", but the remaining of the sentence could not, we think, be much more explicit (?). To our best knowledge the word "rotational" is an official terminology for what can also be called "curl", if that is what is pointed out by the referee. The expression “topologically non-conservative” was intended to refer to a force that is non-conservative, not because it is viscous, but because it is a rotational force field, that is, because of its particular topology. Indeed note that in our system, the mechanical damping rates are unchanged in the presence of the non-conservative force. We propose to modify the expression "topologically non-conservative force fields" into "such rotational force fields".

I do not recommend publishing the paper in Nature Communications.

We thank the referee for his feedback on our work. We hope that the clarification we gave, in particular on the origin of the force, will help the referee changing his opinion on our work.

Reviewer #3 (Remarks to the Author):

The major claims of the paper are finding eigenmode orthogonality breaking, non-reciprocity in eigenmode cross-coupling, an excess of anomalous displacement noise, and a violation of the fluctuation-dissipation theorem.

I advise publication of this manuscript, albeit with major revisions.

We thank the referee for his interest in our work and for his careful review.

The major concern is the length and the focus of the manuscript. Most of the claims were found in reference 16 (eigenmode non-orthogonality, excess of noise, non-reciprocity). The new result here is the analysis of the thermal spectra when the modes become non-orthogonal. This is given consideration, but the findings before the section "Excess of thermal noise" is repetition from reference 16. The other parts of the paper are important for understanding some of the new results, but that should be handled with referencing their 2014 paper (reference 16). Much of the supplementary information is repetitive of the other paper as well, and should be removed and referenced.

We note that in our ref. 16, we were not using a 2D readout but a single 1D projective readout.

This has some important consequences since the eigenmodes orientations changes could not be discriminated from an effective temperature increase. The noise increase mentioned there was only an observation and we could not go beyond this simple statement because it was impossible with a single 1D readout. From this work we know that it is not possible to describe the mechanism by an effective temperature: this would have also been impossible to demonstrate with a single 1D readout channel. The same goes with almost all the other conclusions of the present work.

This is precisely due to this lack of information that we have developed the full 2D readout combined with the spectro-angular noise tomography techniques. It was only at that price that we could correctly conclude on the origin of the observed deviation.

Please note that the force field map of ref 16 was realized at low optical power in order to avoid eigenmode rotation. If there would have been eigenmode rotation, it would have been impossible to extract the force field map: again it would have been impossible to discriminate eigenmode rotation from a change in the force vector orientation with a single 1D readout. At that time we could only give an estimate of the expected eigenmode rotation in presence of the measured gradients (below 3°).

Then, all what concerns eigenmodes rotation/ perpendicularity breaking is an important addition to our previous work and central in the interpretation of our results. It was important for us to show that it can be fully understood.

Given the focus of the manuscript, the referee advises a change in title which emphasizes the new result; that is, something that reflects the change in susceptibility for two cross-coupled non-reciprocal modes. 'Anomalous dynamics' is vague. "Anomalous displacement fluctuations" would be more descriptive.

Not only the fluctuations are "anomalous", the 2D responses as well (larger overall response in presence of eigenmode warping despite the bad orientation of the drive vector with respect to the eigenvectors: Fig 2a), so that we prefer to not restrict to "fluctuations" in the title, if this argument convinces the referee, though we remain open to discussion about the title.

The referee advises removing much of the written material before the analysis of the thermal spectra. The referee also advises moving most of Figure 2 to the supplementary information since it seems to be calibration and not important for the analysis.

The analysis of the eigenmode orthogonality breaking is a central point in the article. In figure 2 we present the results of response and noise measurement analysis: the material presented there is very important in the progression of the article. Once again those are new results that could not have been properly analysed with a single 1D readout. Also, the material presented in Fig. 2 demonstrate the possibility to extract the force field gradients from 2D thermal noise analysis and to compare the results obtained with driven responses. This also extends the demonstration of the suitability of the methodology exposed in our ref. 17 to the case of non-conservative force fields. These points are new and we believe they deserve a clear exposition.

Also we chose to benefit from the long format granted by Nature Communications to write an article as self-consistent as possible.

The referee also calls attention to the additional points:

1) Although it's a small part of this paper, the referee feels the use of the term "topological bifurcation" is confusing. The fact the nanowire follows an ellipse rather than a line after the bifurcation point is immaterial. There is no effect in the spectral data which reflects the change in topology, other than the limit cycle itself. Higher dimensional nonlinear dynamical systems (which this system is given the fact they have negative dissipation past the bifurcation via optomechanical gain, which must be restrained by nonlinear dissipation) show all kinds of multidimensional trajectories (in physical space). The referee does not find this term in the literature of Rayleigh–Bénard convection, where the fluid forms 'topological' patterns upon bifurcation. Using the word 'topological' is confusing and should be removed. The referee advises classifying the bifurcation in accord with the nonlinear dynamics literature.

As pointed out by the referee earlier, the bifurcation and instability that follow have been first observed and studied in a previous article (ref 16) using only a 1D detection. However, in the present work, all of our measurements are made at optical powers below the bifurcation threshold (observed around $85\mu\text{W}$ in the situation of Fig.2). As such the response measurements that are shown in the article are all taken below the instability and in the linear regime. Therefore this work is not focused on this bifurcation. (We also took measurements on the other side of the optical waist, where eigenfrequencies are repelled from each other to demonstrate that the deviation from FDR we analyse is not a consequence of the bifurcation, but of the eigenmode orthogonality breaking.)

In fact, it can be shown with the analytical model that the phenomenology observed here does not depend on the existence of a bifurcation: there exist some force fields (that is, particular configurations of g_{ij} : in the case of a pure shear force: $g_{12}=0$, g_{21} different from 0) where the latter does not happen even at high power, while our conclusions still stand (eigenmode orthogonality breaking, patch of the FDR, etc). One example of such a situation is obtained experimentally on the "right" side of the laser for which we presented analysis in Figure 3 and in the SI. Note that the symmetry is broken between the two sides of the laser by the eigenorientations of the transverse mechanical modes, and that the existence or absence of a bifurcation therefore also depends on these orientations.

We note that in our ref. 16 we made a distinction between *bifurcation* (frequency merging) and *instability* which do not appear at the same optical power. The modification of the damping of both eigenmodes only appears beyond the bifurcation and the eigenvectors also acquire an imaginary part, while the instability appears when the optically underdamped

eigenmode sees its total damping become negative. The instability should result in elliptical trajectories of huge amplitude (see ref16).

Here, we also observe "small" elliptical trajectories (obviously, not as huge as they would be in an instable regime) under a small drive but this is simply a consequence of the fact that both eigenmodes can be simultaneously excited by the driving force due to their quasi-degenerate character (and to the transverse susceptibility contribution at large warping) and they both respond with a different dephasing so that it generates ellipses. The elliptical trajectories are thus not a consequence of the bifurcation or the instability as the oscillator is not in either of these regimes.

For what concerns dissipation, we note that the dissipation of the nanowire does not change in presence of the external force field as long as we stay below the bifurcation threshold, so that all of our spectra are adjusted using the same damping rates (within 5 %, as a consequence of the slight long-term pressure rise in the vacuum chamber where the experiment is performed).

Even if it is not at the origin of the phenomenology exposed here for the reasons previously exposed, the bifurcation is also a consequence of the rotational nature of the force field experienced by the nanowire. As such we use the expression topology of the force field, though we understand the referee's concern and propose to simply remove this adjective.

Finally, since all the physics exposed here appears in the linear regime, it is delicate and probably misleading to try to classify the system according to the literature on non-linear systems. Also up to now we could not find elsewhere similar phenomenology sharing the same physical origin and presenting the same specificities (instantaneous force, non-dissipative, linearity,...).

2)The referee finds the following statement confusing, "It also suggests a very generic methodology to check whether the FDR applies in such a system, that consists in measuring the resemblance between any couple of opposite transverse susceptibilities . This approach therefore only requires the analysis of response measurements that can be repeated and averaged. This remark might be of great use for systems where the experimental verification of the FDR is rendered delicate by a low signal-to-noise ratio that would not allow for the measurement of thermal motion."

How so? Aren't the authors claiming that theirs is the only system with these properties? Maybe they could give a reference for an example.

This statement is connected to the fact that standard verification of the FDR or other fluctuation theorems is often based on statistical analysis of real-time trajectories. This approach is not adequate when the SNR is too low, which is often encountered in optomechanical experiments at low phonon numbers. A method based on response measurements can be time-averaged (or realized with very narrow resolution bandwidth to limit the impact of background noise, and does not suffer from this limitation.

The only uniqueness we claim is the fact of having explored the novel impact of a coupling mechanism that is non-conservative but non-viscous. Since this coupling only originates from the 2D structure of the optomechanical interaction, the observed phenomenology can in principle be observed in any multimode optomechanical system. The effects are enhanced in our case since we work with propagating light beams (on the contrary, optical gradient

(trapping) forces do derive from a potential energy) and with quasi-frequency degenerate oscillators where the effects are enhanced.

We note that multimode *cavity* optomechanical systems are also directly impacted by our observations. In addition, in such systems, due to the finite cavity bandwidth, the corresponding coupling force field could become partially delayed (imaginary part in g_{ij}), and the physics would be even richer, but could still be described by the same 2D coupling matrix.

2) The authors use the word 'immerse'. This is a word for dipping in fluids, not insertion into laser fields (they actually turn on the field around the nanowire so that's not right either). The referee advises a more accurate description of the experimental procedure.

If the referee agrees, we think “Insert” might indeed be more appropriate since we position the green objective to illuminate the nanowire extremity and then adjust its optical power.

3) The authors state "Their ultralow-mass is responsible for a sizeable thermal position noise, that can spread over distances approaching their intrinsic dimensions". The first half of this statement seems incorrect. Isn't position noise proportional to $kT/\text{stiffness}$? Their device has an extreme aspect ratio which would lead to a very low stiffness. Their device actually has a larger mass than typical nanomechanical devices.

The rms position spreading $\text{Sqrt}[kT/\text{stiffness}] = \text{Sqrt}[kT/M\omega^2]$, so yes it would be more appropriate to use ultralow-stiffness (from $1e-7$ to $1e-4$ N/m in our case depending on the geometry).

This sentence was meant to be general and not only associated to our approach/geometry of transverse vibrations for which the word stiffness would have been more adequate (in the case of internal breathing modes, it is always possible to define a stiffness but it becomes less tangible), that is why we employed the more common notion of mass.

We modified the sentence into:

“Their ultralow-mass generally comes with a sizeable thermal position noise...”

Otherwise, we believe the mass of our system to be very reasonably low in the spectrum of oscillators used for force or mass measurement. The only oscillators that are significantly lighter to our knowledge are carbon nanotubes or other kinds of nanotubes, though the motion of these remains considerably more difficult to observe, and optically trapped nano-objects which are also generally difficult to scan to perform force measurement maps. Finally we think the mass of our nanowires is on the same order of magnitude as a very light nano-electro mechanical system (NEMS).

4) The figures are very cramped and difficult to read. The referee suggests removing most of Figure 2 and splitting up or expanding figure 3.

We have tried to render the figures more readable. Following our previous remarks on the importance of Fig. 2, we would like to preserve its structure.

We thank the referee for his feedback on our work.

List of Changes

Abstract:

-Immersion->**insertion**

Introduction

- Their ultralow-mass ... **generally comes with** ... a sizeable thermal position noise modified for clarity
- added : oscillating along both transverse directions (**Fig.\,1a**)
- Removed the reference to Brown1986, the penning trap is sufficiently widely known.
- topologically non-conservative force field replaced by **rotational force field**
- ...radiation pressure force field is **non-conservative** as it presents a non-zero rotational replacing “..is of non conservative nature as...”

The experiment

- while its magnitude, **proportional to the optical pump power**, is varied with an acousto optic modulator... added for clarity.

Intrinsic Nanowire properties

- ... in **the** absence of...
- at even larger drive **amplitudes**
- both fundamental eigenmodes (“longitudinal” removed)

Observation of eigenmodes warping

- immerse replaced by **insert**
- topological instability replaced by to avoid the dynamical instability appearing beyond the bifurcation
- cross-coupled through topologically non-conservative (**rotational**) force fields. added for clarity

Figure 1:

- corrected kHz into Hz in the panel e
- added of non-conservative (**rotational**) nature. for clarity.

Figure 2:

Larger spacing to increase its readability (The definitive figure structure will probably be done / optimized by Nat. Comm.?)

Figure 3:

-Restructured into a 2 column format to improve readability

References

-added a reference to Wu2009 where the rotational optical force field is measured in optical tweezers.

- added a reference to Berry's theoretical work on rotational force field for completeness (Berry2016).

SI

-Added a section describing the calibration and signal analysis routines.

Reviewers' comments:

Reviewer #2 (Remarks to the Author):

The authors have provided a very detailed response to the comments, but made only small changes in the text of the paper to address the concerns in the reports. I do not find the response satisfactory. For example, the argument that the authors are using an approach developed by somebody is not convincing, as the authors cannot explain why this approach is correct and, even if it is correct, why an unconventional approach should be used in their simple and straightforward analysis. The word “topological”, to which two referees objected, is still used, and inappropriately. The modes in the Penning trap remain “nonsymmetrically coupled”, etc.

The statement about the formulation of a “novel modified version of the fluctuation dissipation relation” is an exaggeration, as it implies the generality of the result, which is not the case. Moreover, the central part of the fluctuation dissipation relation is the relation between the intensity of the noise that comes from the coupling to a thermal reservoir and the rate of relaxation due to this coupling. This part is not modified in the considered system, there is nothing new here. The fact that a non-Hermitian matrix has non-orthogonal eigenvectors is known from calculus, so, the non-orthogonality of the modes for a nonpotential force field is hardly an unexpected result. The lifting of the degeneracy of the modes of a wire by rotation is well known in physics, this effect was used to show quantization of vorticity in superfluid helium.

I believe the paper heavily overemphasizes the broad scope and the generality of the results. The use of unconventional terms like “non-axial mechanical susceptibility” complicates the reading. Having said that, the described measurement technique is nice as are also the presented experimental results. I believe the paper deserves to be published provided it is toned down, but I do not see why it should be published in Nature Communications. A more specialized journal would be more appropriate.

Reviewer #3 (Remarks to the Author):

First major point:

The authors do not dispute the fact that the title claims were previously found in ref 16. This is clear since in their supplementary information to ref. 16 they state, "In the case of a conservative force field, the restoring force matrix is symmetric and the eigenvectors are perpendicular to each others. On the contrary, when the force field has a non-conservative character, the eigenvectors are not perpendicular anymore.

Beyond the bifurcation, the eigenvectors acquire an imaginary part, so that the eigen trajectories become elliptical." where the bifurcation was a result of ref 16.

The authors wrote a lengthy response to this point. If the eigenmode orthogonality is important to the manuscript, why is it not mentioned this work is confirmation of a previous result (ref 16) with a 2D measurement? The referee feels the paper is not clearly stating what has been understood before.

The authors ignored the other two items that the referee mentioned (excess of noise, non-reciprocity).

Second Major Point:

"Anomalous" and "Dynamics" are very general terms. These terms could be used in any physical science under any circumstance. The referee believes it is not in the interest of a scientific publication to have these words in the title.

Third Major Point:

At the very least, Figure 2 should be made smaller. It is not necessary to show 7 different values for green laser power. Two should be sufficient to get the point across that the modes orthogonality are breaking. Even though the journal does not have a length requirement, a shorter article without repeating previous results and analysis is easier to read and edit.

1) No comment. Point satisfied.

2) No comment. Point satisfied.

3) The authors with the referee agree that stiffness is correct. It should be changed it to 'compliant'. The thermal forces drive compliant devices to high amplitudes. The manuscript should not propagate incorrect understandings in the field by mentioning mass. The whole key to their experiment is the compliance to force at the nanoscale and not the mass.

165um is very long for a NEMS device. Their effective mass is give as ~1pg. This can be compared to Yang et al 2006 "Zeptogram-Scale Nanomechanical Mass Sensing" which is ~73fg for a SiC device. Their point about being a small mass for NEMS sensors (excepting carbon nanotubes, graphene) is incorrect.

The device in the manuscript is more appropriate for a force sensor than a mass sensor (as has been previously stated). Typically the requirements for force sensors and mass sensors are not complimentary (one being low mass, the other being low stiffness).

4) No comment.

The manuscript is still not ready for publication. Everything about the experiment and it's accompanying analysis is technically correct, but the authors should go beyond cosmetic changes and address the referee points in a thorough manner.

Reviewers' comments:

Reviewer #2 (Remarks to the Author):

The authors have provided a very detailed response to the comments, but made only small changes in the text of the paper to address the concerns in the reports. I do not find the response satisfactory.

For example, the argument that the authors are using an approach developed by somebody is not convincing, as the authors cannot explain why this approach is correct and, even if it is correct, why an unconventional approach should be used in their simple and straightforward analysis.

We thank the referee for his feedback.

The goal of this section in the SI was to calculate the entropy production rate by adapting an approach developed Zamponi et al for a framework very similar to ours, only for more general thermal baths and without spring forces (see Zamponi et al).

The first comment of the referee (response 1) was that equations S75 and S76 (equations number 75 and 76 of the supplementary material) were too sophisticated, and that a double integral was not needed. The equations adapted here happen to be written in the cited reference (Zamponi et al) for a general bath, that is, not generally white and not generally memoryless, in which case the double time integral is needed for the integration of quadratic quantities. In our case of course the correlation of the bath reduces to a delta function due to the assumption of a white, memoryless bath (we added a sentence in the SI section to explicitly state this hypothesis, but notice that for the same system in a liquid this assumption would not generally hold) and our use of their approach was motivated by other similarities between our framework and theirs (same framework but we add spring forces and more specifically spring forces different for each degree of freedom). One of the integrals and the delta function can of course then be removed in eq. S76 and S77 once the hypothesis of the white memoryless bath has been made to write equation S75: we were keeping them despite them being space-consuming because we thought it made it easier to follow the calculation. We have now made the simplification to answer the referee's concern that the equations seemed too sophisticated, which was obviously not our purpose. They now reduce to simple time integrals as the referee was rightly expecting in his first response.

The second comment of the referee (still in response 1) was that path integrals are not needed here. Equation S75 and the following are however not path integrals but a standard stochastic/markovian approach to calculate the thermodynamically relevant quantities, as suggested indeed by the references given by the referee in his first response.

The third comment (still in response 1) was that the calculation should be simple and straightforward and could be found in standard texts. We agree with the referee that this calculation is relatively simple once all hypotheses have been made, however we think that the calculation of the entropy production rate is, to our knowledge, not standard to the point of being found in textbooks as the ones cited, even less so in a rotational force field, as we already stated in our previous response.

To answer the above comment of the referee (response 2) about our response to round 1 of corrections, we are sorry that our reference to the work of Zamponi et al. was interpreted as an authority argument concerning the understanding of the approach. Our purpose was to underline that specialists in the field of thermodynamics start from the *same level of generality* (or even from more general equations) to derive thermodynamics quantities. We thought that the referee was commenting in his first response about equation S75 being too general since it contained a double time integral that could easily be simplified into one. Our response was therefore intended as an (admittedly, authority) argument about format and not substance. We hope that our new answer to his first comment, that now concerns the validity of the approach, will satisfy him.

We would also like to add that we do not claim that our approach is unique.

Finally, those equations S75-S77 are in the SI and more specifically in a section at the end that is only illustrative: they do not play a central role in the article since it only represents an alternative (thermodynamical) analysis of our observations. While we believe it complements our work nicely, we certainly hope that this section is not considered as a major obstacle to the publication of our article. We hope that this answer and the modifications brought to the SI answers all of the referee's concern about this point.

The word “topological”, to which two referees objected, is still used... and inappropriately.

The referee is right, we removed the last « topological » word that remained in the manuscript. We note that referee #3 was satisfied by our response on that aspect.

The modes in the Penning trap remain “nonsymmetrically coupled”, etc.

In our previous response, we explained the reason for the non-reciprocal coupling (through Lorentz force) and removed the reference to Gabrielse paper that was judged inappropriate by the referee.

We have rephrased and completed the sentence to make it clearer.

The statement about the formulation of a “novel modified version of the fluctuation dissipation relation” is an exaggeration, as it implies the generality of the result, which is not the case.

The complete sentence is "a novel modified version of the fluctuation dissipation theorem that remains valid in non-conservative rotational force fields": the precision states the range of the theorem very clearly and with accurate precision in our humble opinion.

However as stated several times in the response and now in the article, the theorem and more generally all of our observations are transposable to non-reciprocally coupled systems that, in fact, share the same mathematical description or have a very similar description as our system subject to rotational forces. This, again, is one of the reasons why we think our analysis of the rotational force system can have an impact far beyond the scope of this article, for example for the dynamic and recent field of non-reciprocal optomechanical systems.

Moreover, the central part of the fluctuation dissipation relation is the relation between the intensity of the noise that comes from the coupling to a thermal reservoir and the rate of relaxation due to this coupling. This part is not modified in the considered system, there is nothing new here.

We believe that we were very clear on that point, which is central in our model and in our demonstration: the violation does not originate from a modification of the Langevin forces but from a transformation by the optical force field of the 2D mechanical susceptibility into a non-reciprocal one.

We have added a sentence at the end of the “model” section state this even more explicitly.

The fact that a non-Hermitian matrix has non-orthogonal eigenvectors is known from calculus, so, the non-orthogonality of the modes for a nonpotential force field is hardly an unexpected result.

In our article, we do not claim that the eigenmode orthogonality breaking is unexpected. However to our best knowledge, it has not been observed before and we believe that we have pushed the analysis far beyond the simple observation of eigenmode orthogonality breaking.

The lifting of the degeneracy of the modes of a wire by rotation is well known in physics, this effect was used to show quantization of vorticity in superfluid helium.

We note that the effects connected to rotation mechanisms involve in general centrifugal forces or similar terms (Coriolis force, etc) which are connected to the speed of the oscillator ($\mathbf{W} \wedge \mathbf{v}$) and cannot be classified in the same category as our system (the coupling force is in $(\text{rot } \mathbf{F}) \wedge \mathbf{r}$).

To further insist on the importance in physics (and the remaining open questions) of this type of force fields (in $\text{rot}(\mathbf{r})$), we have added reference to the recent developments done by M. Berry and colleague on pure $\text{rot}(\mathbf{r})$ forces.

I believe the paper heavily overemphasizes the broad scope and the generality of the results.

We believe that our observations and our conclusions are very general in nano-optomechanics, and can be extrapolated at least to several other systems such as optical tweezers, trapped particles, and systems under active feedback (that can create non reciprocity in multimode devices).

Our system is an ideal test platform, where the different quantities involved take a very simple geometric character and are of easy visualisation, while similar effects may be more complex to extract in other implementations.

The role of the mechanical susceptibility and its tensorial character are central in our observations and cannot be avoided in our analysis.

The use of unconventional terms like “non-axial mechanical susceptibility” complicates the reading.

As can be seen from the definition of the tensorial susceptibility and as is also plainly stated in the text, the "axial susceptibility" characterizes the response of the oscillator in one direction to a force exerted along the same direction, that is, the axial response. The term "non-axial susceptibility" appears just following this definition in the text, and implicitly but -

we think- straightforwardly characterizes the response of the oscillator in a direction other than the direction of the exerted force, that is, other than axial, or non-axial.

We intended not to dwell on this definition because shortly after comes the more useful "transverse susceptibility", also defined in the text straightforwardly as characterizing the response of the oscillator in the direction *perpendicular* to the force exerted on it (if one of the directions, force or displacement, coincides with an eigenmode direction, then this term is zero in conservative force fields, and non-zero in rotational force fields, which we show is at the origin of the FDR violation).

To answer the referee's concern, we however added an explicit definition of the non-axial susceptibility that was only implicit from the definition of the axial susceptibility.

Having said that, the described measurement technique is nice as are also the presented experimental results. I believe the paper deserves to be published provided it is toned down, but I do not see why it should be published in Nature Communications. A more specialized journal would be more appropriate.

We thank the referee for pointing out the interest and the scientific correctness of our work. We have furthermore made an effort to re-evaluate some of the adjectives or complements used to qualify our results a little beyond what the referee explicitly mentioned, see list of changes, but we generally do not think that we have over-emphasized their importance. We thank the referee for his feedback on our manuscript and believe that he will appreciate to see that we have included all of the referee recommendations in our revised version.

Reviewer #3 (Remarks to the Author):

First major point:

The authors do not dispute the fact that the title claims were previously found in ref 16. This is clear since in their supplementary information to ref. 16 they state, "In the case of a conservative force field, the restoring force matrix is symmetric and the eigenvectors are perpendicular to eachothers. On the contrary, when the force field has a non-conservative character, the eigenvectors are not perpendicular anymore.

Beyond the bifurcation, the eigenvectors acquire an imaginary part, so that the eigen trajectories become elliptical." where the bifurcation was a result of ref 16.

The authors wrote a lengthy response to this point. If the eigenmode orthogonality is important to the manuscript, why is it not mentioned this work is confirmation of a previous result (ref 16) with a 2D measurement? The referee feels the paper is not clearly stating what has been understood before.

The authors ignored the other two items that the referee mentioned (excess of noise, non-reciprocity).

Concerning the last sentence: maybe this was unclear in our response, but the requirement for a 2D readout is mandatory not only for investigating and assessing the rotation of eigenmodes, but also for the excess of noise and of course for the non-reciprocity.

However, in absence of 2D readout, nothing can be experimentally proven. In our previous work, which dealt with 1) the experimental demonstration of the existence of rotational optical force fields in focused laser beams through direct measurement 2) the existence characterization and impact of a bifurcation in such a force field and 3) the existence and characterization of a dynamical instability, we only suggested what was predicted by the theory, but this previous article dealt neither with the vectorial aspect of the modes perturbation -since we had no means to measure it- nor with the thermodynamical analysis of the perturbed dynamics in rotational force fields. As such this is not a simple confirmation of previous experimental results. We insist on the importance to have a proper 2D readout, this is not just a technical detail, this is a really important point that should be emphasised throughout this article. Furthermore the regimes explored in ref 16 and in this work are necessarily different since we now concentrate on the physics in a rotational force field *without the existence of a bifurcation* while ref 16 was precisely interested in this phenomenon per se.

Following the referee's suggestion, we have added a sentence in the introduction to state that some of the theoretical developments on eigenmodes orthogonality breaking were already initiated in ref.16, but could not be proven due to the lack of a proper 2D readout.

Furthermore we added a sentence in the beginning section "Observation of eigenmodes warping" at the point where we describe the experimental conditions (and more precisely the insertion into the optical force field) to further precise the distinction between the force regime explored here (avoiding the bifurcation to study other physics) and in our previous work (studying the bifurcation).

Also, as a side note, in ref. 16, we did not analyse the thermal noise spectral shape and magnitude in presence of a large eigenmode rotation since we were puzzled by the strange properties that were expected from the theory (in the first part of the manuscript on the determination of the force field map, the eigenmode rotation was kept very small (low mean optical power), so that the normal mode expansion could be safely be used to fit the response measurements, which is valid at low mean optical power). It took us several years to build a properly calibrated 2D readout that finally allowed us to understand correctly what was behind, and to prove all of the results that are exposed in this work and seriously expand the theoretical analysis.

Second Major Point:

"Anomalous" and "Dynamics" are very general terms. These terms could be used in any physical science under any circumstance. The referee believes it is not in the interest of a scientific publication to have these words in the title.

We were truly surprised by our experimental observations and by the theoretical deduction that could be drawn. This surprise is also a common reaction of our colleagues in the field when we expose our results.

The term "anomalous" has been widely used in the titles of many important articles in several domains (anomalous speed of sound in glasses, anomalous dispersion, ...). Maybe the word "unconventional" could also suit the situation?

We would also like to emphasize the fact that such dynamics as we report might arise "unexpectedly" for example in a multimode optomechanical system (even other than non-reciprocally coupled with a rotational force field) if the equilibrium hypothesis has not been

properly tested -which is the case most of the time since it is indeed quite difficult to test. This is another reason why we would like to underline the contrast with more "usual " situations: we thought of the word anomalous as a warning that strangely-shaped spectra or responses appear very readily in non-reciprocally coupled systems. We are also attached to "dynamics" because "motion" misses the (admittedly slight) nuance that the phenomena exposed here can be transposed to systems other than mechanical or other than involving rotational forces (that is, systems that are just non-reciprocally coupled). We suggest that the editor could take the final decision on this point.

Third Major Point:

At the very least, Figure 2 should be made smaller. It is not necessary to show 7 different values for green laser power. Two should be sufficient to get the point across that the modes orthogonality are breaking. Even though the journal does not have a length requirement, a shorter article without repeating previous results and analysis is easier to read and edit.

We sincerely apologize that due to a file exchange in our revised manuscript we failed to submit our latest version of Fig.2 with our last response, in which we had indeed already reduced the number of items from 7 to 4 in Fig 2a.

We agree with the referee that this indeed helps increasing the readability of this figure.

However, as exposed above, we do not agree on the statement that this part is a repetition of previous results.

Concerning Figure 2, every part of it is discussed in the text and is or leads to a direct, important and new experimental result: the direct observation of warping on Fig 2a which also details all vectors orientations that are needed in the rest of the text to interpret the physics, the anomalous lineshapes of Fig 2b, the deduced frequencies and modes orientations of Fig 2c and 2d which show a merging of the orientations at the same time as the merging of the frequencies (it also serves as a more standard way to represent data than Fig 2a, and details the data points that were removed on Fig 2a as suggested), the g_{ij} deduced from these on Fig 2e as an illustration of the non-zero force rotational and of an imperfection of our system (departure from linearity) that we discuss in the text, and the measured noise excess in the modes directions recomputed from the data analysis on Fig 2f that illustrates equation 2. We insist that none of these results are calibration steps. Concerning the manuscript itself, we conclude this answer (see further) with an explanation as to why we still believe that it should not be shortened.

1) No comment. Point satisfied.

We have removed all the "topological" terms in the manuscript

2) No comment. Point satisfied.

We have added "in real time" to be even clearer on that point. We have clarified even more the concerned sentence.

3) The authors with the referee agree that stiffness is correct. It should be changed it to

'complaint'. The thermal forces drive compliant devices to high amplitudes. The manuscript should not propagate incorrect understandings in the field by mentioning mass. The whole key to their experiment is the compliance to force at the nanoscale and not the mass.

165um is very long for a NEMS device. Their effective mass is give as ~1pg. This can be compared to Yang et al 2006 "Zeptogram-Scale Nanomechanical Mass Sensing" which is ~73fg for a SiC device. Their point about being a small mass for NEMS sensors (excepting carbon nanotubes, graphene) is incorrect.

The device in the manuscript is more appropriate for a force sensor than a mass sensor (as has been previously stated). Typically the requirements for force sensors and mass sensors are not complimentary (one being low mass, the other being low stiffness).

The referee is right, the rms spatial spreading of the thermal noise only depends on the oscillator stiffness, or on its compliance, so we have followed his recommendation and modified our sentence accordingly.

However, let us remark that if one considers the thermal noise as the ultimate limit for force sensing (such as when one desires measuring an AC force), then the stiffness (or compliance) is not the only relevant parameter to mention: the Langevin force noise spectral density is given by $S_F = 2 M W m k_B T/Q$ cannot be simply expressed only using the temperature and the oscillator's stiffness ($k \propto M W m^2$). The problem may be different for probing force gradient, especially when active driving is used, but in general, the force sensitivity cannot be only reduced to the oscillator stiffness since it also depends on the Q factor, which cannot be thought to be completely independent from the geometry (mass or stiffness) at the nanoscale.

4) No comment.

We have now modified figures 2 and 3 to follow the referee comments on their readability.

The manuscript is still not ready for publication. Everything about the experiment and it's accompanying analysis is technically correct, but the authors should go beyond cosmetic changes and address the referee points in a thorough manner.

To facilitate the readability of our work, in particular towards readers that do not have a full subscription to other journals, we think it is preferable to have one readable manuscript almost self-consistent instead of a shorter version that would require multiple bibliographic references. We believe that this is also in agreement with the open access policy provided by Nat. Comm. but we also propose that the editor takes the final decision on that point.

We also point out that those are only the first paragraphs at the beginning of the sections "The experiment" and "Intrinsic nanowire properties" which are not fully novel but describe the experiment and establish a context which is important for the progression of the article.

Besides this suggested reduction of the manuscript content, on which we do not agree for pedagogic reasons and not to remove novel results as exposed above, we think that we have already modified the manuscript to fulfil all the referee comments (title, vocabulary, figures, etc). Since we cannot not find major modifications to be done beyond that and since the

referee agrees that the technical and scientific content is correct we hope that this version will now satisfy the referee.

We thank the referee again for his feedback on our work.

List of changes

Figure 2

Modified version, including a reduced number of powers in 2a.

Introduction

- Their ultralow mass...=> Their ultralow **stiffness**...
- Added: "It was theoretically suggested that it could generate a warping of the eigenmode basis and an altered dynamics, but no experimental conclusion could be drawn since only a scalar 1D readout was available at that time." (the importance of the 2D readout is explained later in the manuscript)
- Modified+ Added: **...if a charged particle is immersed in a magnetic field, the components of its motion along perpendicular directions are non-symmetrically coupled through the Lorentz force:: the force experienced along the two transverse directions being proportional to the speed along the other direction but with an opposite sign.**
- -added :"**We show experimentally that** the position fluctuations now deviate from the fluctuation dissipation relation (FDR) "
- Modified: "The field of cavity optomechanics **has recently developed a strong interest in** multimode coupling phenomena such as..."
- Modified :"**..., in a simple system** where all discussed quantities can be directly mapped and visualized in a 2D space."

The experiment

- Added : "Three particular positions within the force-applying laser, visible on Fig 1d, were investigated, each featuring a different local force structure: the central position denoted as \otimes as a zero-rotational test point and the positions \circlearrowright and \circlearrowleft for positive and negative force rotational."

Intrinsic nanowire properties

- Modified: "**Firstly, these** were determined..."
- Modified : "The displacement associated with each mode is **rectilinear along direction forming** angles of respectively.... so that **the modes** are found to be..."

Observation of eigenmode warping

- added “We now focus on the vectorial aspect of the nanowire motion perturbed by rotational forces. We insist that this work deals with rotational forces that *do not* entail a dynamical bifurcation as in \cite{Gloppe2014}. To do so we keep the optical powers below the onset threshold or work with optical force field gradient structures $\{g_{ij}\}$ that will not cause a bifurcation at any optical power.”

-suppression of “topologically” in: “... at play in 2D oscillators cross-coupled through *topologically* non-conservative (rotational) force fields.”

Model

- changed S_F into S_F^{th} for clarity

-added “The **cosine** angle between eigenvectors follows...”

-Added: “As in standard cavity optomechanics, our model makes use of a mechanical susceptibility modified by the light field (which can become non-reciprocal when the coupling force field presents a non-zero rotational) while the Langevin forces and intrinsic damping rates remain unchanged.”

Anomalous thermal noise spectra

-added :”Such a deviation may **in some systems** originate from...”

-added “...but deviate from the expected linearity **at large optical powers.**”

Deviation from the fluctuation dissipation theorem

-added “A similar control measurement set was realized in absence of non-conservative force field **in position \otimes** using...”

-modified :”... also loose their orthogonality but **eigenfrequencies** are repelled from each other instead of merging.”

-added “This underlines that the deviation is not due to the frequency merging mechanism but instead - **as will now be discussed** - to the eigenmodes orthogonality breaking...”

Non-axial contributions

-added :” **For the purpose of illustration**, we focus on...”

-added “is now sought in the non-axial susceptibility terms, **that is, the component that characterizes the response in one direction to a force exerted along another direction.**”

Modified fluctuation dissipation relation

-modified “Further developing the model exposed above, it is possible to demonstrate, see SI, that the deviation from FDR observed in **such a non-reciprocally coupled system** verifies...”

- displaced :” We note that the quadratic dependence of the deviation in $\text{rm } \mathbf{v}\{\text{rot}\}\mathbf{v}\{F\}$ originates from the fact that the NW is exploring a rotational force field along Brownian trajectories whose statistics is unbalanced by the non-conservative force (see SI).”

-modified “On the contrary, **driving along** $\mathbf{v}\{e_{-}^{\wedge}\text{perp}\}\approx \mathbf{v}\{e_{-}F\}$, **produces a** significant displacement along...”

- modified “...by damping forces (see SI), **according to** the Harada-Sasa theorem $\text{cite}\{\text{Harada2005, Harada2006}\}$ for **any** out-of-equilibrium systems.”

-Added: This approach therefore only requires the analysis of response measurements that can be repeated and averaged **and realized with a small resolution bandwidth in order not to be limited by the measurement background noise.**

- added : ” In contrast with the original theorem, we propose here a spectral, non-integrated formulation of the deviation to the FDR **applied to non-reciprocally coupled systems**”

- added: This remark might be of great use for systems where the experimental verification of the FDR is rendered delicate by a low signal-to-noise ratio that would not allow for the measurement of thermal motion **in real time.**

- modified “Hence this equation **is important** for the description of fluctuations...” (to tone down the article.)

Conclusion

-added a reference to Berry, M. V. & Shukla, P. Curl force dynamics: symmetries, chaos and constants of motion, *New Journal of Physics*, **2016**, 18, 063018.

-modified : « This system may also be of great interest to test **new** formulations of fluctuation theorems... »

- added : « It is also an interesting platform for investigating real time dynamics in rotational force fields $\text{cite}\{\text{Berry2016}\}$. »

Supplementary material

-modified S_F into $S_{F^{\wedge}\text{rm th}}$

Thermodynamical aspects

-added after equation S75 : “...where $\delta(t-t')$ is used here because of our assumption that the thermal bath is memoryless and delta-correlated.”

-re-expressed equations **S76** and **S77** without using the dirac term.

SNR

-removed the section, useless here.

Former List of Changes

Abstract:

-Immersion->**insertion**

Introduction

- Their ultralow-mass ... **generally comes with** ... a sizeable thermal position noise modified for clarity
- added : oscillating along both transverse directions (**Fig.\,1a**)
- Removed the reference to Brown1986, the penning trap is sufficiently widely known.
- topologically non-conservative force field replaced by **rotational force field**
- ...radiation pressure force field is **non-conservative** as it presents a non-zero rotational replacing "...is of non conservative nature as..."

The experiment

- while its magnitude, **proportional to the optical pump power**, is varied with an acousto optic modulator... added for clarity.

Intrinsic Nanowire properties

- ... in **the** absence of...
- at even larger drive **amplitudes**
- both fundamental eigenmodes ("longitudinal" removed)

Observation of eigenmodes warping

- immerse replaced by **insert**
- topological instability replaced by to avoid the dynamical instability appearing beyond the bifurcation
- cross-coupled through topologically non-conservative (**rotational**) force fields. added for clarity

Figure 1:

- corrected kHz into Hz in the panel e
- added of non-conservative (**rotational**) nature. for clarity.

Figure 2:

Larger spacing to increase its readability (The definitive figure structure will probably be done / optimized by Nat. Comm.?)

Figure 3:

-Restructured into a 2 column format to improve readability

References

-added a reference to Wu2009 where the rotational optical force field is measured in optical tweezers.
- added a reference to Berry's theoretical work on rotational force field for completeness (Berry2016).

SI

-Added a section describing the calibration and signal analysis routines.